# RED: Robust Event-Guided Motion Deblurring with Modality-Specific Disentangled Representation

## Abstract

Event cameras provide sparse yet temporally high-resolution motion information, demonstrating great potential for motion deblurring. However, the delicate events are highly susceptible to noise. Although noise can be reduced by raising the threshold of Dynamic Vision Sensors (DVS), this inevitably causes under-reporting of events. Most existing event-guided deblurring methods overlook this practical trade-off. The modality-indiscriminate feature extraction and naive fusion treat images and events as statistically similar inputs, leading to unstable and mixed representations, especially when events are disrupted. To tackle these challenges, we propose a Robust Event-guided Deblurring (RED) network with modality-specific disentangled representation. First, we introduce a Robustness-Oriented Perturbation Strategy (RPS) that mimics various DVS thresholds, exposing RED to diverse under-reporting patterns and thereby fostering robustness under unknown conditions. To better exploit partially disrupted events, we design a Modality-specific Representation Mechanism (MRM) that disentangles the inputs into three complementary components: an image-semantic representation capturing structure and textures, an event-motion representation extracting fine-grained motion details, and a cross-modal representation modeling complementary interactions. Building on these reliable features, two interactive modules are presented to enhance motion-sensitive areas in blurry images and inject semantic context into under-reporting event representations. Extensive experiments on synthetic and real-world datasets demonstrate RED consistently achieves state-of-the-art performance in terms of both accuracy and robustness.

## 1 Introduction

Motion blur is a pervasive degradation in dynamic visual scenes, typically caused by rapid object motion or unintended camera shake during exposure. Image deblurring aims to reconstruct sharp textures and structures from a blurred observation, with approaches spanning handcrafted priors (Levin et al., 2009; Krishnan et al., 2011; Bahat et al., 2017) to modern CNN- and Transformer-based models (Nah et al., 2017b; Tao et al., 2018; Dong et al., 2023; Kong et al., 2023; Liang et al., 2024; Wang et al., 2024). Yet under severe motion blur, critical structural and temporal details are heavily corrupted, and the performance of these methods degrades markedly.

Event cameras, inspired by biological vision systems, have emerged as a promising alternative to conventional sensors in high-speed scenes (Gallego et al., 2020). They emit asynchronous streams of events that encode motion with high temporal precision, making them naturally well suited for motion deblurring. Early methods (Pan et al., 2019; Zhang & Yu, 2022) relied on physical models to establish deterministic mappings between blurred images and events, whereas recent studies have adopted deep learning-based solutions that enable more expressive and powerful representations. In particular, cross-modal fusion strategies such as event-image attention mechanisms (Sun et al., 2022), spatio-temporal attention modules (Yang et al., 2024), and frequency-aware interactions (Kim et al., 2024) have been introduced to bridge the gap between events and blurry images. Furthermore, feature interaction strategies such as unidirectional event-guided fusion (Sun et al., 2023), bidirectional modality alignment (Chen & Yu, 2024), and asymmetric bidirectional integration (Yang et al., 2025) further explore the synergistic potential of multi-modal features.

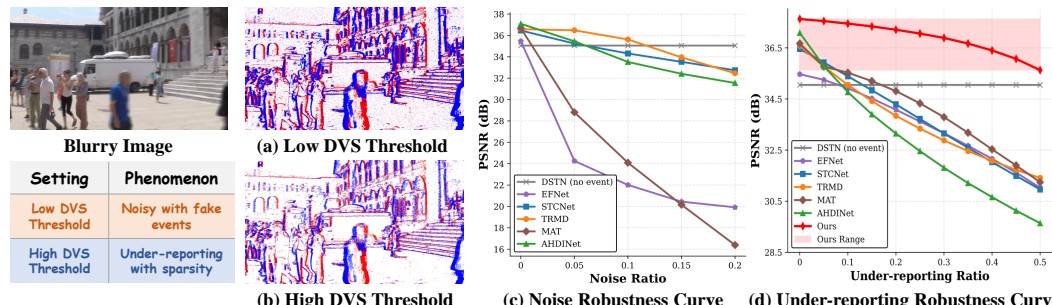

**Figure 1: Robustness analysis.** Performance of existing event-based deblurring methods to noisy events and under-reporting events individually under low and high DVS threshold are shown in (c) and (d). As the noise or under-reporting ratio increases, the performance degrade sharply and fall below the image-only method DSTN.

Despite these advances, event cameras are intrinsically susceptible to noise because an event is triggered whenever the log-intensity change at a pixel exceeds a contrast threshold. This threshold-triggering mechanism introduces various noise sources, such as background activity, photon shot noise, and electronic circuit noise, which carries little semantic information and contaminates the event stream (Guo & Delbruck, 2022; Duan, 2024). As shown in Figure 1c, the performance of recent event-guided deblurring models drops sharply as the noise ratio increases. Training with noisy events can partially improve robustness under heavy noise, but this inevitably sacrifices fidelity due to the low quality of events and the difficulty of extracting reliable motion priors.

Alternatively, raising the threshold of Dynamic Vision Sensors (DVS) can suppress spurious noise events to some extent (Vishenvskiy et al., 2024). However, this inevitably induces threshold-driven under-reporting as weak motions and low-contrast edges fall below the trigger condition are not reported. As illustrated in Figure 1d, existing methods tolerate mild under-reporting but degrade rapidly once the ratio exceeds 0.20, even performing worse than image-only deblurring (Pan et al., 2023). This contradicts the common assumption that events could monotonically provide positive contributions to deblurring. Under pronounced under-reporting, the marginal utility of events becomes non-positive and can even be detrimental. We attribute this to the fact that current pipelines ignores the disrupted events in practice, and often process images and events indiscriminately with shared feature extractors or naive fusion, leading to unstable and mixed representations and ultimately unsatisfactory results. This motivates the need for a modality-specific representation that isolates event-specific motion evidence while preserving image structures, followed by robustness-aware cross-modal interactions to adaptively leverage reliable features.

In this paper, we present a Robust Event-guided Deblurring network (RED) with modality-specific disentangled representation. First, we introduce a Robustness-Oriented Perturbation Strategy (RPS), which mimics realistic under-reporting events by varying DVS thresholds, enabling the network to adapt to diverse event dropouts. It simultaneously expose key issues of deblurring network in practice: indiscriminate extraction or naive fusion introduces unreliable features, and the absence of weak events makes deblurring more difficult. To address these issues and effectively adapt to RPS, a Modality-specific Representation Mechanism (MRM) is designed to disentangle semantic, motion, and cross-modality features. By separating semantic information from images and motion details from events before fusion, MRM prevents corrupted event characteristics from overwhelming image semantics and allows our RED to retain useful motion priors. Finally, two modules are designed to achieve coadjutant interactions: a Motion Saliency Enhancer Module (MSEM) that transfers motion-sensitive priors to enhance spatial details easily lost in blur, and an Event Semantic Engraver Module (ESEM) that delineate semantic characteristics from images into deeper event encoding, mitigating the semantic deficiency caused by sparse events. In this way, RED achieves reliable deblurring through robust event adaptability, disentangled representation learning, and effective cross-modal interaction. In summary, our main contributions are as follows:

1. We propose RED for robust event-guided motion deblurring. Extensive experiments demonstrate our RED outperforms existing methods in both deblurring quality and robustness on synthetic and real-world datasets.

2. RPS exposes our RED to diverse under-reporting events in various DVS thresholds, enhancing the robustness and adaptability to real-world conditions.

3. MRM is tailored to factorize feature space into semantic and temporal dimensions for disentangled modality-specific representation, while robust motion-sensitive priors and compensatory semantic context are interacted by MSEM and ESEM.

## 2 RELATED WORK

### 2.1 IMAGE DEBLURRING

Image deblurring seeks to recover a clear image from its motion-blurred counterpart, a degradation typically caused by camera shake or dynamic object motion. Classical image deblurring methods typically follow a two-step pipeline: estimating the blur kernel and performing non-blind deconvolution (Fergus et al., 2006; Cho & Lee, 2009). To constrain this ill-posed problem, various image priors, such as Total Variation (Chan & Wong, 1998), sparsity (Xu et al., 2013), patch recurrence (Michaeli & Irani, 2014), and color statistics(Pan et al., 2016) have been explored. While effective in controlled settings, these approaches struggle with spatially-varying blur and rely heavily on accurate kernel modeling (Tran et al., 2021; Hyun Kim et al., 2013), often leading to unstable performance and high computational cost. With the advent of deep learning, CNN-based architectures have demonstrated notable advances by directly learning the blur-to-sharp mapping from large-scale images. Multi-scale feature fusion (Nah et al., 2017b; Tao et al., 2018; Dong et al., 2023), multi-stage refinement (Chen et al., 2021; Zamir et al., 2021b; Yang et al., 2022; Liu et al., 2024), and coarse-to-fine strategies (Cho et al., 2021; Zheng et al., 2022) have been extensively explored. More recently, attention-based models leveraging both spatial- or frequency- domain representations (Tsai et al., 2022; Zamir et al., 2022; Kong et al., 2023; Mao et al., 2024) have achieved state-of-the-art performance by capturing long-range dependencies and preserving fine-grained structures. Despite these advancements, existing models still struggle under severe or complex blur conditions.

### 2.2 EVENT-BASED MOTION DEBLURRING

Event cameras offer low-latency sensing by asynchronously recording per-pixel brightness changes, making them well-suited for high-speed motion deblurring. Early works such as EDI (Pan et al., 2019) attempted to explicitly model the physical relationship, while recent methods have leveraged deep learning to construct more robust and expressive representations. A straightforward approach concatenated event with RGB images to assist image deblurring (Jiang et al., 2020), but such naive integration failed to capture fine-grained interactions across modalities. D2Nets (Shang et al., 2021) designed a learnable weight matrix to embed event priors into arbitrary image deblurring networks. Further advancing event representation, EFNet (Sun et al., 2022) introduced a symmetric cumulative event encoding combined with a cross-attention mechanism to enhance inter-modal feature interaction. To better exploit modality complementarity and suppress redundancy, STCNet (Yang et al., 2024) designed a cross-modal co-attention module that dynamically fuses spatial features from both streams. FEVD (Kim et al., 2024) presented a frequency-domain filtering approach that captures global cross-modal dependencies via spatial frequency interactions. Most recently, AHDINet (Yang et al., 2025) argued that symmetric or unidirectional fusion strategies (Sun et al., 2023; Chen & Yu, 2024) may lead to either insufficient or redundant interactions. They instead proposed an asymmetric bidirectional integration framework to more effectively leverage complementary information.

Despite these advances, most existing methods implicitly assume that event streams are complete and reliable. In practice, however, events are inherently noisy or under-reporting due to the thresholding mechanism of DVS, and this disrupts the quality of motion information. With an ignorance of the challenging events in practice, shared feature extractors or generic fusion strategies in current pipelines fail to effectively disentangle semantic context from images and motion evidence from practical events, leading to fragile cross-modal representations and reduced robustness.

## 3 METHODS

In this paper, we propose a Robust Event-guided Deblurring (RED) network in Figure 2. In detail, RED first performs Robustness-Oriented Perturbation Strategy (RPS) on event input and then extract modality-specific chracteristics via Modality-specific Representation Mechanism (MRM) ,

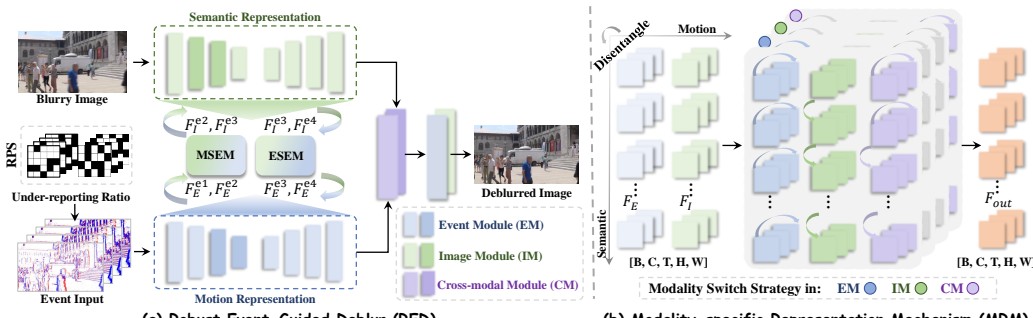

(a) Robust Event-Guided Deblur (RED)   (b) Modality-specific Representation Mechanism (MRM)

Figure 2: **Overview of our RED.** In detail, a Robustness-Oriented Perturbation Strategy (RPS) is implemented to event input, and MRM is designed to disentangle modality-specific features with individually semantic reasoning in image module, motion-wise representation in motion module, and cross-modality fusion in cross-modal module. Furthermore, Motion Saliency Enhancer Module (MSEM) is designed to excavate motion-sensitive priors to image branch and Event Semantic Engraver Module (ESEM) is presented to compensate events with global semantic understanding.

cooperating with two interactive modules, Motion Saliency Enhancer Module (MSEM) and Event Semantic Engraver Module (ESEM) for complementary modality interaction.

## 3.1 ROBUSTNESS-ORIENTED PERTURBATION STRATEGY (RPS)

Event cameras generate an event when the logarithmic intensity change at a pixel exceeds a contrast threshold $\theta \in \mathbb{R}_+$. Formally, let the log-intensity increment at pixel coordinates $(x, y)$ and time $t$ be

$$\Delta \ell(x, y, t) = \log I_t(x, y) - \log I_{t-\Delta t}(x, y), \tag{1}$$

where $I_t(x, y)$ denotes the intensity at time $t$, and $\Delta t$ is the time interval. We decompose $\Delta \ell$ into a signal and a noise component:

$$\Delta \ell = S + N, \tag{2}$$

where $S$ corresponds to true motion signals and $N$ accounts for noise. An event is triggered when $|\Delta \ell| \geq \theta, p = \text{sign}(\Delta \ell)$, which highlights a fundamental trade-off between sensity and noise. When $S$ is weak, the events are easily dominated by noise $N$, making the stream highly susceptible to corruption, shown in Figure 1c and details in ***Appendix*** **A.2**.

In practice, event noise originates from multiple physical sources (Jiang et al., 2024), such as photon shot noise and thermal or electronic fluctuations, which can be modeled as Poisson and Gaussian process. To capture in a unified way, we represent the aggregated noise as a mixed distribution

$$N = N_p + N_g, \qquad N_p \sim \text{Poisson}(\lambda), \qquad N_g \sim \mathcal{N}(0, \sigma_n^2), \tag{3}$$

where $N_p$ accounts for photon arrivals with rate $\lambda$, and $N_g$ represents circuit-induced perturbations with variance $\sigma_n^2$. An event is incorrectly triggered when $|N_p + N_g| \geq \theta$. The corresponding false positive rate (FPR) with $\mathbb{P}(\cdot)$ as the probability of an event is

$$\text{FPR}(\theta) = \mathbb{P}(|N_p + N_g| \geq \theta), \tag{4}$$

which decreases as $\theta$ increases. When true motion $S$ is present, the true positive rate (TPR) is

$$\text{TPR}(\theta \mid S) = \mathbb{P}(|S + N_p + N_g| \geq \theta \mid S). \tag{5}$$

Although larger thresholds reduce FPR, they also decrease TPR by discarding weak but informative events. We quantify this by the under-reporting ratio (UR):

$$\text{UR}(\theta \mid S) = 1 - \text{TPR}(\theta \mid S), \tag{6}$$

whose expectation over the distribution of $S$ defines the overall under-reporting:

$$\text{UR}(\theta) = \mathbb{E}_S[\text{UR}(\theta \mid S)]. \tag{7}$$

In summary, $\theta$ acts as a global control knob: $\theta \uparrow \Rightarrow$ FPR $\downarrow$, UR $\uparrow$. This motivates our RPS, which explicitly reformulates event acquisition as a probabilistic triggering process. At pixel $(x, y)$ and time $t$, whether an event survives is determined by the survival probability:

$$\pi_t(x, y) = \mathbb{P}\big(|S + N_p + N_g| \geq \theta\big), \tag{8}$$

which encodes the likelihood that the log-intensity increment surpasses the contrast threshold. Thus, the problem naturally reduces to a binary decision, an event either survives or is suppressed.

To emulate this mechanism during training, we adopt the voxel grid representation $\mathbf{D} \in \mathbb{R}^{T \times H \times W}$ of events (Sun et al., 2022), where $T$ denotes the number of temporal bins. At each slice $\tau \in \{1, \ldots, T\}$, we stochastically thin the events according to their survival probability:

$$\widetilde{\mathbf{D}}_\tau = \mathbf{D}_\tau \odot \boldsymbol{\rho}_\tau, \qquad \boldsymbol{\rho}_\tau(x, y) \sim \text{Bernoulli}\big(\pi_\tau(x, y)\big), \tag{9}$$

where $\odot$ denotes element-wise multiplication. During training, we further use UR to control perturbation strength. In detail, for each iteration we sample

$$\alpha \sim \mathcal{U}(\alpha_{\min}, \alpha_{\max}), \qquad \frac{1}{HW} \sum_{x,y} \pi_\tau(x, y) \approx 1 - \alpha, \quad \forall \tau, \tag{10}$$

and draw $\boldsymbol{\rho}_\tau$ independently across temporal bins to mimic variability in thresholds and circuitry. This generates a continuum of training regimes from mild to severe under-reporting, faithfully reflecting the physical mechanism of threshold-driven event triggering. As verified in Tables 1, 7 and Figure 9, our architecture-agnostic and parameter-free RPS substantially improves the robustness of RED, ensuring high-quality deblurring under diverse scenes.

### 3.2 Modality-specific Representation Mechanism (MRM)

Blurry images primarily encode high-level semantic context, whereas over-reporting events provide complementary motion priors but lack complete semantic information. When both modalities are indiscriminatively extracted or naively processed together, the lack of disentangling can cause semantic and motion features to mix, reducing the clarity of the extracted representations. Thus, we rethink effective event-based image deblurring with modality-specific representations, where MRM individually perform semantic-wise, motion-wise, and cross-modality representation.

**Semantic-wise and Motion-wise Attention.** Given the intermediate features $\mathbf{F}_{\text{I}}^{ei}$ and $\mathbf{F}_{\text{E}}^{ei} \in \mathbb{R}^{B \times N \times H \times W}$ from the image and event branches, where $B$ denotes the batch size and $N$ refers to the original mixed space. To emphasize semantic understanding, we begin by disentangling the feature space. Using a $1 \times 1$ convolution followed by a depthwise $3 \times 3$ convolution, we obtain the query, key, and value tensors for semantic modeling:

$$\mathbf{Q}_{\text{sem}}, \mathbf{K}_{\text{sem}}, \mathbf{V}_{\text{sem}} \in \mathbb{R}^{B \times L \times C_L \times (THW)}, \tag{11}$$

where $N = C \times T$ represents the split between the channel and temporal dimensions, and $C = C_L \times L$ denotes the multi-head splitting, with $L$ being the number of heads. For motion-specific modeling, tokens are reshaped as $\mathbf{Q}_{\text{mot}}, \mathbf{K}_{\text{mot}}, \mathbf{V}_{\text{mot}} \in \mathbb{R}^{B \times L \times T_L \times (CHW)}$, where $T = L \times T_L$. Here, the splitting focuses on temporal indices, ensuring the attention explicitly captures motion continuity across time. Thus, the corresponding semantic correlation and motion dependency are formulated as

$$\mathbf{A}_{\text{sem}} = \text{Softmax}(\mathbf{Q}_{\text{sem}} \mathbf{K}_{\text{sem}}^\top), \quad \mathbf{A}_{\text{mot}} = \text{Softmax}(\mathbf{Q}_{\text{mot}} \mathbf{K}_{\text{mot}}^\top). \tag{12}$$

Then, the updated features are then computed as

$$\mathbf{F}_{\text{I}}^{ei} = \mathbf{V}_{\text{sem}} \odot \mathbf{A}_{\text{sem}}, \quad \mathbf{F}_{\text{E}}^{ei} = \mathbf{V}_{\text{mot}} \odot \mathbf{A}_{\text{mot}}. \tag{13}$$

In this way, semantic characteristics are enhanced in the image branch, while temporal motion dependencies are emphasized in the event branch.

**Cross-Modality Attention.** With the former modality-specific representation, a cross-modality interaction are designed to broadcast semantic understanding and motion changes. Considering the event modality lacks global semantic structure under high under-reporting ratios, whereas the image modality still provides stable and complete semantic priors. On the one hand, when transferring semantic cues from images to events, both the query and key are taken from the image modality to construct a reliable semantic attention map. In detail, query and key are derived from $\mathbf{F}_{\text{I}}^{ei}$,

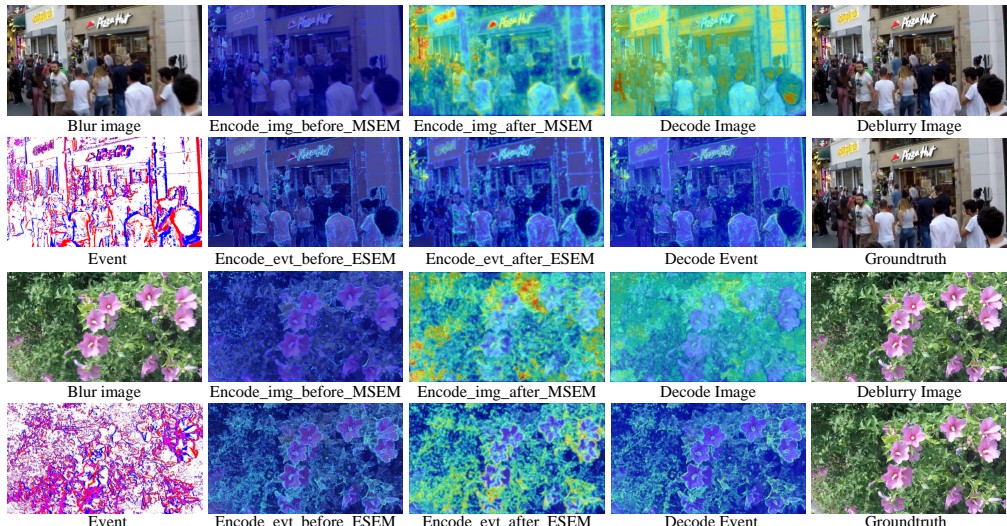

| Blur image | Encode_img_before_MSEM | Encode_img_after_MSEM | Decode Image | Deblurry Image |
| Event | Encode_evt_before_ESEM | Encode_evt_after_ESEM | Decode Event | Groundtruth |
| Blur image | Encode_img_before_MSEM | Encode_img_after_MSEM | Decode Image | Deblurry Image |
| Event | Encode_evt_before_ESEM | Encode_evt_after_ESEM | Decode Event | Groundtruth |

Figure 3: **Visualizations.** In detail, we visualize the activation maps, including: 1) image-encoder features before and after MSEM, 2) image-branch decoder features, 3) event-encoder features before and after ESEM, and 4) event-branch decoder features.

while value is taken from $\mathbf{F}_{\mathrm{E}}^{ei}$. Here, spatio-temporal tokens are chosen, thus features from images attributes to delineating abundant semantic understanding.

$$\mathbf{A}_{I \to E} = \mathrm{Softmax}\big(\mathbf{Q}_I \mathbf{K}_I^{\top}\big), \qquad \widetilde{\mathbf{F}}_E = \mathbf{V}_E \odot \mathbf{A}_{I \to E}. \tag{14}$$

On the other hand, when transferring motion details from events to images, both the query and key are taken from the event modality to construct a reliable motion dependency. In detail, query and key are derived from $\mathbf{F}_{\mathrm{E}}^{ei}$, while value comes from $\mathbf{F}_{\mathrm{I}}^{ei}$. Here, spatio-channel tokens are chosen, thus events guides the images to restore detailed structural distributions.

$$\mathbf{A}_{E \to I} = \mathrm{Softmax}\big(\mathbf{Q}_E \mathbf{K}_E^{\top}\big), \qquad \widetilde{\mathbf{F}}_I = \mathbf{V}_I \odot \mathbf{A}_{E \to I}. \tag{15}$$

Finally, two-mode outputs are concatenated to obtain the fused output $\mathbf{F}_{\mathrm{out}}$. In conclusion, MRM switches among semantic-wise, motion-wise, and cross-modality attentions, seperately serving as the core mechanism of image, event, and cross-modal modules. By this way, MRM embody our RED with modality-specific representations and effective cross-modality fusion.

### 3.3 MSEM AND ESEM

In our whole RED framework in Figure 2, Motion Saliency Enhancer Module (MSEM) and Event Semantic Engraver Module (ESEM) are designed to achieve coadjutant interactions. As illustrated in Figure 4, MSEM aims to highlight motion-sensitive structures from events and inject them into the image branch. Given the perturbed event input $\widetilde{\mathbf{D}}$, we first derive high-frequency components $\mathbf{S} \in \mathbb{R}^{B \times N \times H \times W}$ via a downsample-then-subtract operation followed by depthwise convolutions.

The motion-enhanced event feature is then initialized as:
$$\widehat{\mathbf{F}}_{\mathrm{E}}^{(1)} = \mathbf{F}_{\mathrm{E}}^{(i)} \odot \mathbf{S} + \mathbf{F}_{\mathrm{E}}^{(i)},$$

where $\odot$ denotes element-wise multiplication. To transfer motion priors into the image branch, $\widehat{\mathbf{F}}_{\mathrm{E}}^{(1)}$ is concatenated with $\mathbf{F}_{\mathrm{I}}^{(i)}$ and processed by a $1 \times 1$ convolution followed by two depthwise convolutions, yielding a coarse fused representation $\mathbf{F}_{\mathrm{mix}}$. From $\mathbf{F}_{\mathrm{mix}}$, we compute a motion-aware attention map

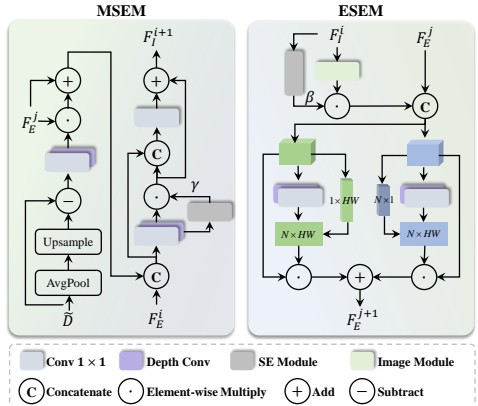

Figure 4: Framework of our proposed Motion Saliency Enhancer Module (MSEM) and Event Semantic Engraver Module (ESEM).

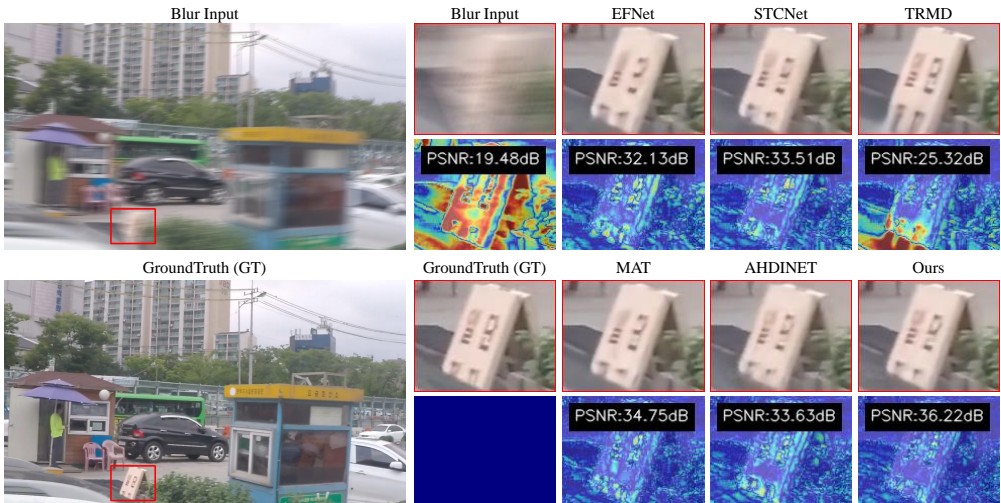

Figure 5: **Visualization in GoPro dataset.** Besides, error maps are drawn to illustrate a comprehensive comparison of both localization and structure context.

$\gamma \in \mathbb{R}^{B \times N \times H \times W}$. Using this map, the event feature is refined as: $\widehat{\mathbf{F}}_{\mathrm{E}}^{(2)} = \gamma \odot \widehat{\mathbf{F}}_{\mathrm{E}}^{(1)}$. Finally, $\mathbf{F}_{\mathrm{mix}}$ is concatenated with $\widehat{\mathbf{F}}_{\mathrm{E}}^{(2)}$ to produce a fine-grained motion-enhanced feature $\mathbf{F}_{\mathrm{E2I}}$, which is then propagated to update the next-stage image representation $\mathbf{F}_{\mathrm{I}}^{(i+1)}$. As illustrated in Fig. 3, after MSEM, the image branch preserves its semantic structure, but exhibits enhanced responses exactly at event-indicated motion areas.

Complementary to MSEM, ESEM engraves high-level semantic representations from the image branch into the motion branch. Given $\mathbf{F}_{\mathrm{I}}^{(i)}$, we first extract a latent spatial feature $\widehat{\mathbf{F}}_{\mathrm{I}}^{(i)}$ through an image encoder. A channel-wise attention then generates correlation weights $\boldsymbol{\beta}$, and the semantic embedding is formulated as: $\mathbf{F}_{\mathrm{sem}}^{(i)} = \boldsymbol{\beta} \odot \widehat{\mathbf{F}}_{\mathrm{I}}^{(i)}$. To enhance event features with this semantic prior, $\mathbf{F}_{\mathrm{sem}}^{(i)}$ is fused with $\mathbf{F}_{\mathrm{E}}^{(i)}$ to form $\widehat{\mathbf{F}}_{\mathrm{mix}}$. We split $\widehat{\mathbf{F}}_{\mathrm{mix}}$ along the channel dimension into $\widehat{\mathbf{F}}_{\mathrm{mix}}^{\mathrm{E}}$ and $\widehat{\mathbf{F}}_{\mathrm{mix}}^{\mathrm{sem}}$, which are individually processed by a dual-branch attention module: temporal attention on $\widehat{\mathbf{F}}_{\mathrm{mix}}^{\mathrm{E}}$ produces an event-aware output $\overline{\mathbf{F}}_{\mathrm{E}}^{(i)}$, while spatial attention on $\widehat{\mathbf{F}}_{\mathrm{mix}}^{\mathrm{sem}}$ produces a spatially modulated feature $\overline{\mathbf{F}}_{\mathrm{sem}}^{(i)}$. The two outputs are concatenated to yield the final semantically enriched event feature $\mathbf{F}_{\mathrm{I2E}}$, which is forwarded to the next-stage event representation $\mathbf{F}_{\mathrm{E}}^{(i+1)}$. As illustrated in Fig. 3, after ESEM, the event branch receives semantic reinforcement from the image branch, producing motion representations with more complete contours and improved structural continuity.

## 4 EXPERIMENTS

### 4.1 DATASETS

We evaluate our method on both synthetic and real-world event-based deblurring datasets. Specifically, we adopt the widely used GoPro dataset (Nah et al., 2017a), which provides pairs of blurry and sharp images synthesized by averaging high-speed frames. GoPro contains 3,214 image pairs at a resolution of $1280 \times 720$, including 2,103 pairs for training and 1,111 for testing. To further assess the robustness, we leverage 412 blurry images of size $1632 \times 1224$ from the HighREV dataset (Sun et al., 2023), along with their corresponding event streams. Besides, six sequences from the REVD dataset (Kim et al., 2024) are adopted, which includes 1,359 blurry images captured in typical urban scenes with diverse motion patterns, such as ego-motion, object motion, and their combinations.

### 4.2 IMPLEMENTATION DETAILS

In the training process, $256 \times 256$ patches are cropped, and 400k iterations are conducted during the training process. Besides, Equation 10 is set as $\alpha \sim \mathcal{U}(\alpha_{\min} = 0, \alpha_{\max} = 0.2)$. We employ the Adam optimizer with an initial learning rate of $2 \times 10^{-4}$, $(\beta_1, \beta_2) = (0.9, 0.999)$ and $\epsilon = 10^{-8}$. To

Table 1: **Comparisons on GoPro dataset.** Metrics are PSNR (↑) and SSIM (↑). UR stands for the various under-reporting ratio on events. DSTN denotes image deblurring without event assistance.

| UR | DSTN | EFNet | STCNet | TRMD | AHDINet | MAT | Ours |
|---|---|---|---|---|---|---|---|
| 0 | 35.05 / 0.973 | 35.47 / 0.972 | 36.46 / 0.975 | 36.58 / 0.979 | 37.09 / 0.978 | 36.67 / 0.978 | **37.63 / 0.9802** |
| 0.05 | 35.05 / 0.973 | 35.25 / 0.971 | 35.93 / 0.973 | 35.77 / 0.976 | 35.78 / 0.973 | 35.77 / 0.974 | **37.55 / 0.9800** |
| 0.1 | 35.05 / 0.973 | 34.92 / 0.970 | 35.39 / 0.972 | 35.05 / 0.973 | 34.77 / 0.969 | 35.53 / 0.973 | **37.45 / 0.9797** |
| 0.15 | 35.05 / 0.973 | 34.52 / 0.968 | 34.84 / 0.970 | 34.42 / 0.970 | 33.90 / 0.964 | 35.20 / 0.972 | **37.34 / 0.9794** |
| 0.2 | 35.05 / 0.973 | 34.08 / 0.966 | 34.29 / 0.967 | 33.84 / 0.967 | 33.15 / 0.960 | 34.81 / 0.970 | **37.21 / 0.9790** |
| 0.3 | 35.05 / 0.973 | 33.16 / 0.960 | 33.15 / 0.962 | 32.88 / 0.961 | 31.81 / 0.951 | 33.79 / 0.965 | **36.89 / 0.9781** |

maintain the training stability, a gradual warmup strategy in (Zamir et al., 2021a) is applied in our training process with a minimum learning rate of $1 \times 10^{-6}$. The Peak Signal-to-Noise Ratio (PSNR) and the Structural Similarity Index Metric (SSIM) are adopted as the evaluation metrics.

### 4.3 COMPARABLE RESULTS

Five state-of-the-art event-based image deblurring methods are chosen as fair and straight comparisons, including EFNet (Sun et al., 2022), STCNet (Yang et al., 2024), TRMD(Chen & Yu, 2024), AHDINet (Yang et al., 2025), and MAT (Xu et al., 2025). Besides, DSTN (Pan et al., 2023) as an image deblurring method, is adopted as a reference to verify the effectiveness of introducing events.

**Comparable Results in GoPro:** Table 1 reports quantitative comparisons on the GoPro dataset. Across different under-reporting ratios (UR), RED consistently achieves the highest PSNR and SSIM. As shown in Figure 6, existing event-based methods exhibit a sharp performance drop as UR increases, indicating their inability to extract effective motion priors from over-reporting events. This finding echoes our motivation: simply injecting event inputs without considering modality-specific characteristics renders models highly vulnerable to event

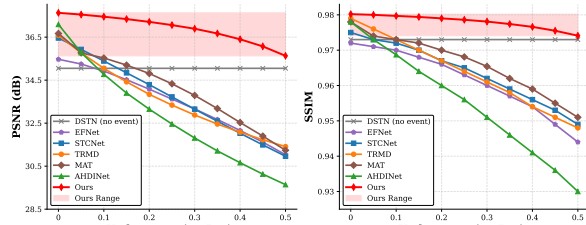

Figure 6: **Performance comparisons.** Our RED maintains a stable capablity to promote deblurring performance even under severe under-reporting events.

corruption. In contrast, RED maintains stable performance even at UR = 0.5, outperforming the image-only baseline. This robustness is largely attributed to the diverse under-reporting patterns introduced by our RPS module, as well as the MRM, which disentangles modality-specific features and enables efficient cross-modality interactions. Furthermore, visual results in Figure 5 demonstrate that our method produces sharper details and cleaner textures. *More detailed numerical results and performance analysis are presented in **Appendix** A.5, while additional qualitative results are provided in **Appendix** A.7.*

**Comparable Results in HighREV and REVD:** To further verify the generalization of event-based methods in practice, we conduct deblurring experiments on HighREV and REVD datasets with the former trained models on GoPro. As shown in Tables 2a and 2b, our method consistently achieves the highest PSNR and SSIM, demonstrating strong robustness across diverse scenes and motion patterns. Besides, visual comparisons on HighREV dataset in Figure 7 and challenging REVD dataset in Figure 8 show Ours achieves clearer results. Interestingly, we observe that PSNR of deblurred image in some methods is lower than that of blurry input. This phenomenon indicates their failure to extract useful motion priors from real-world events for deblurring, further validating the motivation behind our approach. *More qualitative results are presented in **Appendix** A.7.*

### 4.4 ABLATION STUDY

**Investigation of RPS.** To evaluate the effectiveness of our RPS, we first validate RED and RED without RPS under different UR settings in Figure 9a. Then, we integrate RPS into existing event-based methods to further assess its efficiency, including Transformer-based MAT (Xu et al., 2025) in

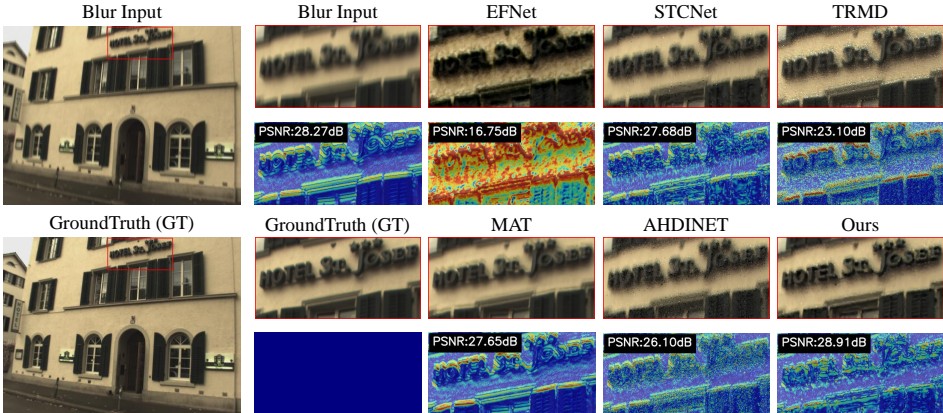

Figure 7: **Visualization in HighREV dataset.** Ours demonstrates more clear text edge and cleaner background.

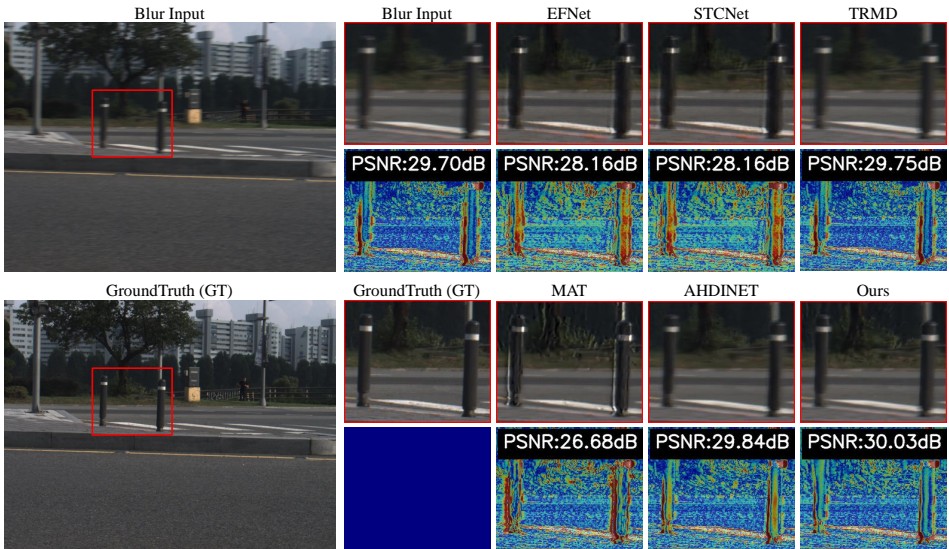

Figure 8: **Visualization in REVD dataset.** Ours reconstructs more continuous contours, clearer grass textures, and cleaner edge transitions.

Table 2: **Quantitative comparisons.** Metrics are PSNR (↑) and SSIM (↑). Best results on HighREV (*Left*) and REVD (*Right*) datasets are in **bold**.

| Methods | EFNet | STCNet | TRMD | MAT | AHDINet | Ours |
|---------|-------|--------|------|-----|---------|------|
| PSNR(↑) | 20.56 | 28.77 | 28.89 | 27.54 | 28.82 | **30.04** |
| SSIM(↑) | 0.808 | 0.869 | 0.892 | 0.922 | 0.857 | **0.929** |

(a) Comparisons on HighREV dataset.

| Methods | EFNet | STCNet | TRMD | MAT | AHDINet | Ours |
|---------|-------|--------|------|-----|---------|------|
| PSNR(↑) | 26.17 | 26.97 | 26.88 | 24.83 | 26.30 | **27.35** |
| SSIM(↑) | 0.829 | 0.857 | 0.858 | 0.803 | 0.822 | **0.860** |

(b) Comparisons on REVD dataset.

Figure 9b and CNN-based AHDINet (Yang et al., 2025) in Figure 9c. Finally, an overall comparison in Figure 9d is conducted on Ours, MAT with our RPS, and AHDINet with our RPS.

From Figure 9, the following conclusions are dramn. First, removing RPS from our framework leads to significant robustness drops. Second, incorporating RPS into existing event-guided methods consistently improves their robustness, demonstrating the plug-and-play generality of RPS. Finally, beyond the contribution of RPS, our method consistently achieves the best performance and robustness across all conditions, highlighting the importance of our modality-specific representation.

To further verify the efficiency of our RPS, as formulated in Equ. 9 and 10, the FLOPs $= TCHW + 3TCHW \cdot DR = TCHW(1 + 3DR)$, where $T$ is the number of event frames, $(C, H, W)$ are the channel and spatial dimensions, and $DR$ denotes the disrupted ratio. To further quantify the

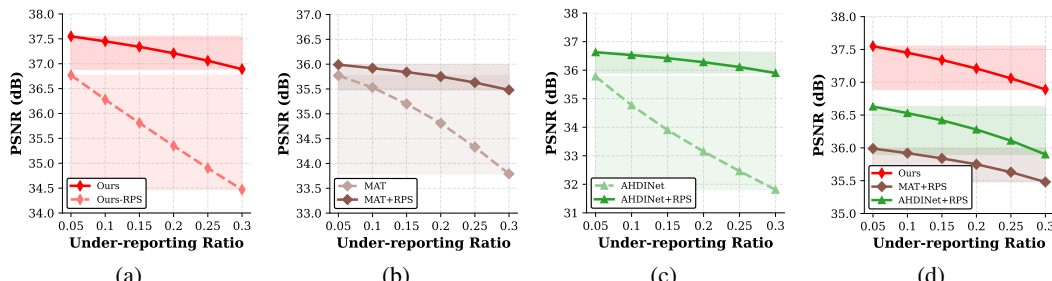

(a)   (b)   (c)   (d)

Figure 9: **Verification on our RPS.** We report results of (a) Ours v.s. Ours w/o RPS, (b) MAT v.s. MAT+RPS, (c) AHDINet v.s. AHDINet+RPS, and (d) Ours v.s. MAT+RPS/AHDINet+RPS.

overhead, we test the runtime of RPS over 100 runs on an event tensor of size $(B, T, C, H, W) = (1, 6, 1, 360, 640)$. RPS introduces only 2.49M additional FLOPs and roughly 0.71 ms of runtime overhead when applied to the above tensor.

**Investigation of MRM.** To assess the contribution of different components in MRM, we conduct a comprehensive ablation study by selectively replacing the semantic reasoning, motion-wise structural representation, and cross-modality attention mechanisms with a modality-agnostic self-attention module (Yuan et al., 2021), while keeping the feature dimensionality unchanged. As shown in Table 3a, replacing all specialized attentions with generic self-attention causes a dramatic performance drop of 11.86 dB in PSNR, indicating that indiscriminately processing blurry images and events severely impairs effective feature extraction and interaction. Replacing only the modality-specific attention in the image or event branch results in 2.64dB and 2.59dB drops, while a smaller but noticeable degradation of 0.49dB when replacing the cross-modality attention. These findings collectively demonstrate that our MRM, not only captures the distinct semantic context and motion-sensitive areas, but also facilitates more effective and complementary feature fusion.

Table 3: **Ablation study**. Metrics are PSNR ($\uparrow$) and SSIM ($\uparrow$). Best results are in **bold**.

| Semantic | Motion | Cross | PSNR($\uparrow$) | SSIM($\uparrow$) |
|:---:|:---:|:---:|:---:|:---:|
| ✗ | ✗ | ✗ | 25.77 | 0.864 |
| ✗ | ✓ | ✓ | 34.99 | 0.964 |
| ✓ | ✗ | ✓ | 35.04 | 0.966 |
| ✓ | ✓ | ✗ | 37.14 | 0.978 |
| ✓ | ✓ | ✓ | **37.63** | **0.980** |

(a) Ablation study on MRM.

| MSEM | ESEM | PSNR($\uparrow$) | SSIM($\uparrow$) |
|:---:|:---:|:---:|:---:|
| ✗ | ✗ | 36.78 | 0.976 |
| ✗ | ✓ | 36.97 | 0.977 |
| ✓ | ✗ | 37.18 | 0.978 |
| ✓ | ✓ | **37.63** | **0.980** |

(b) Ablation study on MSEM and ESEM.

**Investigation of MSEM and ESEM.** We further investigate the contributions of two feature interaction modules in our RED. As reported in Table 3b, each module individually improves PSNR and SSIM over the baseline. The joint cooperation of MSEM and ESEM leads to the best performance, with a 0.85dB gain in PSNR over the baseline. Besides, we observe that the performance gain from feature interaction strategies is less pronounced than that from modality-specific attention in Table 3a. This also supports our hypothesis when deigning RED: modality-specific feature encoding benefits to modality-agnostic representation and enables more effective cross-modality collaboration, ultimately leading to superior deblurring performance.

## 5 CONCLUSION

In this paper, we present RED, a robust and practical event-based deblurring framework that explicitly accounts for the under-reporting of event data caused by the trade-off between sensitivity and noise. We introduce a RPS to embody RED with powerful and robust adaptability under different unknown scenario conditions. To extract fine-grained motion priors from under-reporting events and prevent corrupted effect, a disentangled MRM is designed to explicitly model semantic, motion, and cross-modality correlations, allowing more effective extraction of modality-specific features and cross-modality fusion. With the robust event representation and modality-specific disentangled representation, our RED delivers state-of-the-art deblurring performance and strong generalization.

## ETHICS STATEMENT

This work relies on publicly available datasets under their respective licenses. No new data involving human subjects were collected. All visualizations respect privacy. We confirm that our method and experiments do not raise additional ethical concerns

## REPRODUCIBILITY STATEMENT

We use publicly accessible datasets, GoPro (Nah et al., 2017a), HighREV (Kim et al., 2024), and REVD (Sun et al., 2023). After the blind review period, we will release our training/inference scripts, configuration files, and model checkpoints, together with step-by-step instructions and evaluation protocols to fully reproduce all tables and figures.

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

# A APPENDIX

In the appendix, we first provide the LLM usage statement. Then, we analyze the impact of low and high DVS thresholds on the performance of event-based deblurring methods, and then illustrate the process we focus on extracting effective motion priors from more clean events under high DVS threshold. Next, we demonstrate the activation maps of corresponding features in our RED. Subsequently, we illustrate the detailed structure of our modality-interaction module: MSEM and ESEM. Next, we provide a more detailed analysis between the model performance and efficiency. Then, more visualizations on simulated and real-world datasets are presented. Finally, we demonstrate the results to further support the generalization of RPS to other sensor degradations.

**Appendix is organized as follows:**

CONTENTS

## A.1 USE OF LLMs

The LLMs are used only for language polishing and editing of the manuscript text.

## A.2 ROBUSTNESS ANALYSIS OF EXISTING METHODS UNDER LOW AND HIGH DVS THROSHOLD

Here, we investigate the performance of existing event-based deblurring methods with events in low DVS threshold and high DVS threshold in various practical scenes. Event cameras generate asynchronous events when the change of log-intensity at a pixel exceeds a contrast threshold $\theta$. As discussed in the main paper, this mechanism inherently involves a trade-off:

1) Low threshold ($\theta \downarrow$): The sensor becomes highly sensitive but produces many spurious events dominated by noise.

2) High threshold ($\theta \uparrow$): Noise can be suppressed but weak and informative events fall below the threshold, resulting in under-reporting.

Thus, the quality of the event stream depends critically on $\theta$, which controls the balance between noise contamination and information loss. Comprehensive robustness verification experiments have been conducted on existing deblurring methods, which are illustrated in Figure 10a and Figure 10b.

**Robustness of Existing Methods under Noise (Low Threshold).** To study robustness against noise, we simulate different noise ratios under low threshold settings. Table 4 reports PSNR/SSIM values for state-of-the-art event-based deblurring methods and the image-only baseline (DSTN). It can be observed that:

1) When the noise ratio is small (0.05), most event-based methods already degrade notably compared to the clean case.

2) As the noise ratio increases ($\geq 0.15$), existing methods collapse severely, falling even below the image-only baseline.

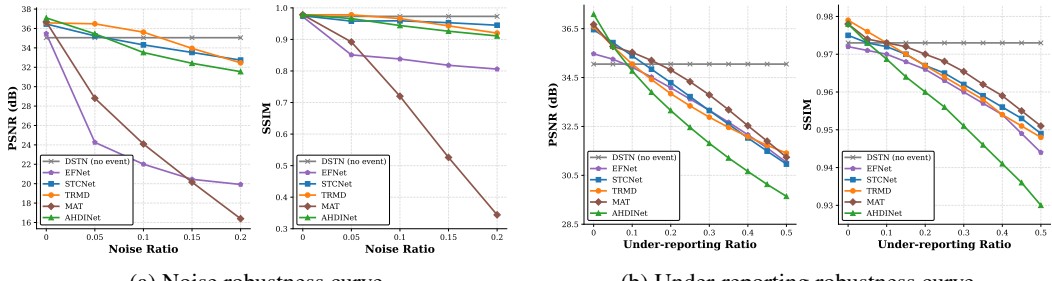

(a) Noise robustness curve.
(b) Under-reporting robustness curve.

Figure 10: **Visualization of robustness comparisons in Table 4 and Table 5.** As the level of noise and under-reporting ratio increases, the (PSNR↑ / SSIM↑) of event-based methods degrade sharply and often fall below the image-only baseline DSTN. It indicates that current pipelines fails a robust capability to extract motion priors disrupted events. Relatively, accurate motion information is more critical for deblurring than merely noisy events. This motivates our RPS: instead of relying on idealized events, RPS explicitly exposes the model to threshold-induced disruptions during training, enabling RED to adapt to diverse real-world scenes.

Table 4: **Performance under low DVS theshold.** Robustness of existing event-based deblurring methods under different noise perturbance ratios. DSTN denotes image-only deblurring baseline.

| Noise Ratio | DSTN (no event) | EFNet | STCNet | TRMD | AHDINet | MAT |
|---|---|---|---|---|---|---|
| 0 | 35.05 / 0.973 | 35.47 / 0.972 | 36.46 / 0.975 | 36.58 / 0.979 | 37.09 / 0.978 | 36.67 / 0.978 |
| 0.05 | 35.05 / 0.973 | 24.26 / 0.851 | 35.24 / 0.958 | 36.48 / 0.978 | 35.45 / 0.966 | 28.81 / 0.892 |
| 0.1 | 35.05 / 0.973 | 22.01 / 0.838 | 34.31 / 0.959 | 35.62 / 0.966 | 33.52 / 0.944 | 24.09 / 0.720 |
| 0.15 | 35.05 / 0.973 | 20.45 / 0.818 | 33.53 / 0.953 | 33.95 / 0.943 | 32.41 / 0.926 | 20.16 / 0.526 |
| 0.2 | 35.05 / 0.973 | 19.92 / 0.806 | 32.74 / 0.945 | 32.46 / 0.920 | 31.55 / 0.911 | 16.39 / 0.344 |
| 0.3 | 35.05 / 0.973 | 19.92 / 0.785 | 31.35 / 0.925 | 30.02 / 0.871 | 30.26 / 0.881 | 15.07 / 0.281 |

Table 5: **Performance under high DVS theshold.** Robustness of existing event-based deblurring methods under different under-reporting ratios. DSTN denotes image-only deblurring baseline.

| UR | DSTN (no event) | EFNet | STCNet | TRMD | AHDINet | MAT |
|---|---|---|---|---|---|---|
| 0 | 35.05 / 0.973 | 35.47 / 0.972 | 36.46 / 0.975 | 36.58 / 0.979 | 37.09 / 0.978 | 36.67 / 0.978 |
| 0.05 | 35.05 / 0.973 | 35.25 / 0.971 | 35.93 / 0.973 | 35.77 / 0.976 | 35.78 / 0.973 | 35.77 / 0.974 |
| 0.1 | 35.05 / 0.973 | 34.92 / 0.970 | 35.39 / 0.972 | 35.05 / 0.973 | 34.77 / 0.969 | 35.53 / 0.973 |
| 0.15 | 35.05 / 0.973 | 34.52 / 0.968 | 34.84 / 0.970 | 34.42 / 0.970 | 33.90 / 0.964 | 35.20 / 0.972 |
| 0.2 | 35.05 / 0.973 | 34.08 / 0.966 | 34.29 / 0.967 | 33.84 / 0.967 | 33.15 / 0.960 | 34.81 / 0.970 |
| 0.25 | 35.05 / 0.973 | 33.62 / 0.963 | 33.72 / 0.965 | 33.34 / 0.964 | 32.46 / 0.956 | 34.33 / 0.968 |
| 0.3 | 35.05 / 0.973 | 33.16 / 0.960 | 33.15 / 0.962 | 32.88 / 0.961 | 31.81 / 0.951 | 33.79 / 0.965 |
| 0.35 | 35.05 / 0.973 | 32.67 / 0.957 | 32.59 / 0.959 | 32.46 / 0.958 | 31.21 / 0.946 | 33.18 / 0.962 |
| 0.4 | 35.05 / 0.973 | 32.16 / 0.954 | 32.03 / 0.956 | 32.08 / 0.954 | 30.66 / 0.941 | 32.53 / 0.959 |
| 0.45 | 35.05 / 0.973 | 31.62 / 0.949 | 31.49 / 0.953 | 31.73 / 0.951 | 30.13 / 0.936 | 31.90 / 0.955 |
| 0.5 | 35.05 / 0.973 | 31.03 / 0.944 | 30.96 / 0.949 | 31.41 / 0.948 | 29.64 / 0.930 | 31.24 / 0.951 |

**Robustness of Existing Methods under Under-Reporting (High Threshold).** Next, we evaluate robustness to incomplete events caused by high thresholds. Table 5 reports PSNR/SSIM under different under-reporting ratios (UR). It can be observed that:

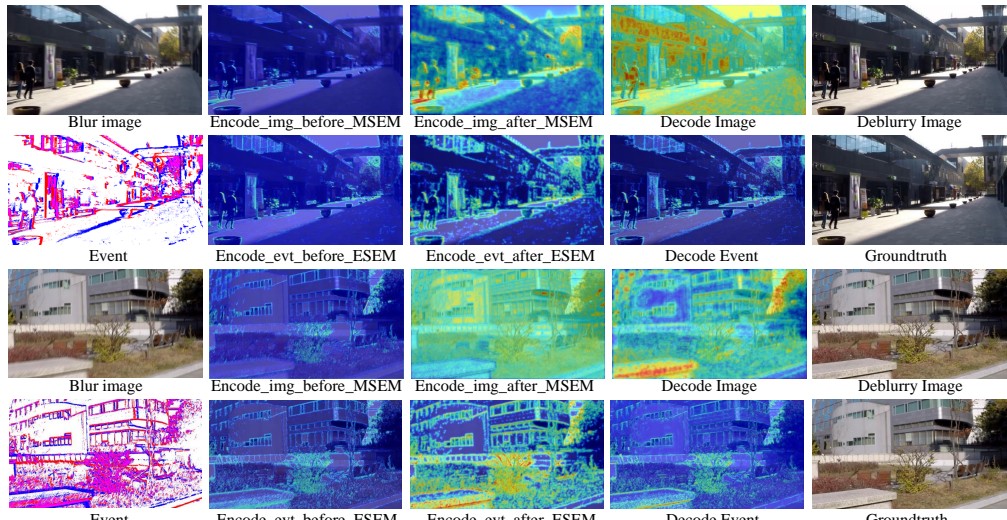

Figure 11: **Visualizations.** In detail, we visualize the activation maps, including: 1) image-encoder features before and after MSEM, 2) image-branch decoder features, 3) event-encoder features before and after ESEM, and 4) event-branch decoder features.

1) Performance degrades progressively as UR increases. While mild under-reporting (UR $\leq 0.1$) is tolerable, all event-based methods show sharp declines once UR $\geq 0.2$.

2) Notably, the performance eventually drops below the image-only baseline (DSTN), which contradicts the common assumption that events always provide positive contributions.

**Key Insight and Motivation of RPS.**    The above analysis of quantitative results in Tables 4, 5 and corresponding qualitative results in Figure 10, it can be deduced that accurate motion information is more critical for deblurring than merely noisy events. This motivates our RPS: instead of relying on idealized events, RPS explicitly exposes the model to threshold-induced disruptions during training, enabling RED to adapt to diverse real-world scenes.

## A.3    VISUALIZATION OF ACTIVATION MAPS

Besides, we visualize the activation maps in Figs. 3 and 11. We can infer that: 1) Before MSEM, image activations exhibit broad semantic layouts, while event activations focus on high-motion regions. 2) After MSEM, the image branch preserves its semantic structure, but exhibits enhanced responses exactly at event-indicated motion areas. 3) After ESEM, the event branch receives semantic reinforcement from the image branch, producing motion representations with more complete contours and improved structural continuity. 4) In the decoder, the image modality consistently governs global structure and coherence, while the event modality strengthens local detail recovery and edge transitions. These observations demonstrate that the primary functional separation between semantic and motion cues remains intact throughout the pipeline, and that MSEM/ESEM operate as intended: providing directed cross-modal compensation without collapsing the disentangled representation space.

## A.4    DETAILED STRUCTURE OF OUR MSEM AND ESEM

In our whole RED framework in Figure 2, Motion Saliency Enhancer Module (MSEM) and Event Semantic Engraver Module (ESEM) are designed to achieve coadjutant interactions. As illustrated in Figure 12, MSEM aims to highlight motion-sensitive structures from events and inject them into the image branch. Given the perturbed event input $\widetilde{\mathbf{D}}$, we first derive high-frequency components $\mathbf{S} \in \mathbb{R}^{B \times N \times H \times W}$ via a downsample-then-subtract operation followed by depthwise convolutions.

The motion-enhanced event feature is then initialized as:

$$\widehat{\mathbf{F}}_{\mathrm{E}}^{(1)} = \mathbf{F}_{\mathrm{E}}^{(i)} \odot \mathbf{S} + \mathbf{F}_{\mathrm{E}}^{(i)},$$

where $\odot$ denotes element-wise multiplication. To transfer motion priors into the image branch, $\widehat{\mathbf{F}}_{\mathrm{E}}^{(1)}$ is concatenated with $\mathbf{F}_{\mathrm{I}}^{(i)}$ and processed by a $1\times1$ convolution followed by two depthwise convolutions, yielding a coarse fused representation $\mathbf{F}_{\mathrm{mix}}$. From $\mathbf{F}_{\mathrm{mix}}$, we compute a motion-aware attention map $\boldsymbol{\gamma} \in \mathbb{R}^{B \times N \times H \times W}$. Using this map, the event feature is refined as:

$$\widehat{\mathbf{F}}_{\mathrm{E}}^{(2)} = \boldsymbol{\gamma} \odot \widehat{\mathbf{F}}_{\mathrm{E}}^{(1)}.$$

Finally, $\mathbf{F}_{\mathrm{mix}}$ is concatenated with $\widehat{\mathbf{F}}_{\mathrm{E}}^{(2)}$ to produce a fine-grained motion-enhanced feature $\mathbf{F}_{\mathrm{E2I}}$, which is then propagated to update the next-stage image representation $\mathbf{F}_{\mathrm{I}}^{(i+1)}$.

Complementary to MSEM, ESEM engraves high-level semantic representations from the image branch into the motion branch. Given $\mathbf{F}_{\mathrm{I}}^{(i)}$, we first extract a latent spatial feature $\widehat{\mathbf{F}}_{\mathrm{I}}^{(i)}$ through an image encoder. A channel-wise attention then generates correlation weights $\boldsymbol{\beta}$, and the semantic embedding is formulated as: $\mathbf{F}_{\mathrm{sem}}^{(i)} = \boldsymbol{\beta} \odot \widehat{\mathbf{F}}_{\mathrm{I}}^{(i)}$. To enhance event features with this semantic prior, $\mathbf{F}_{\mathrm{sem}}^{(i)}$ is fused with $\mathbf{F}_{\mathrm{E}}^{(i)}$ to form $\widehat{\mathbf{F}}_{\mathrm{mix}}$. We split $\widehat{\mathbf{F}}_{\mathrm{mix}}$ along the channel dimension into $\widehat{\mathbf{F}}_{\mathrm{mix}}^{\mathrm{E}}$ and $\widehat{\mathbf{F}}_{\mathrm{mix}}^{\mathrm{sem}}$, which are individually processed by a dual-branch attention module: temporal attention on $\widehat{\mathbf{F}}_{\mathrm{mix}}^{\mathrm{E}}$ produces an event-aware output $\overline{\mathbf{F}}_{\mathrm{E}}^{(i)}$, while spatial attention on $\widehat{\mathbf{F}}_{\mathrm{mix}}^{\mathrm{sem}}$ produces a spatially modulated feature $\overline{\mathbf{F}}_{\mathrm{sem}}^{(i)}$. The two outputs are concatenated to yield the final semantically enriched event feature $\mathbf{F}_{\mathrm{I2E}}$, which is forwarded to the next-stage event representation $\mathbf{F}_{\mathrm{E}}^{(i+1)}$.

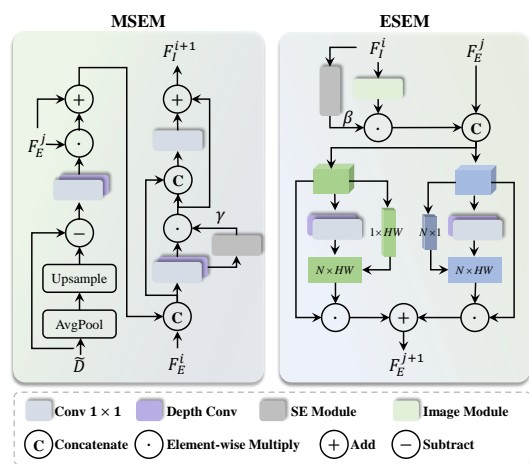

Figure 12: Framework of our proposed Motion Saliency Enhancer Module (MSEM) and Event Semantic Engraver Module (ESEM).

## A.5 PARAMETERS AND FLOPS

As illustrated in Table 6, we present a comparison between model complexity and performance, with a fixed spatial resolution of $360 \times 640$. In particular, EFNet, STCNet, and AHDINet adopt CNN-based architectures, enhanced by several cross-attention mechanisms to fuse event and blurry image features. These models generally maintain moderate parameter counts and computational costs, but varying performance under increasing UR. For example, while AHDINet achieves strong results under clean events, its performance drops significantly as UR increases, indicating limited temporal robustness.

On the other hand, TRMD, MAT, and our proposed method employ Transformer-based architectures as the backbone, resulting in higher parameter counts compared to the CNN-based approaches. This is expected, as Transformer modules are typically more expressive but computationally intensive. Notably, our method achieves the best overall performance across all conditions, attaining 37.63 dB PSNR at UR = 0 and maintaining 35.63 dB even at UR = 0.5, while keeping the parameter count lower than MAT and comparable to TRMD.

These results highlight the trade-off between efficiency and robustness. While CNN-based methods are lightweight, their temporal generalization tends to be limited. Despite being computationally heavier, Transformer-based designs provide superior robustness to event disturbances. Our proposed method achieves a favorable balance between these two aspects, validating the effectiveness of its architecture in handling diverse motion scenes.

Table 6: **Comparison between model complexity and performance on GoPro dataset.** Metrics are PSNR (↑) and SSIM (↑). UR stands for the various under-reporting ratio on events. DSTN denotes image deblurring without event assistance.

| UR | DSTN | EFNet | STCNet | TRMD | AHDINet | MAT | Ours |
|---|---|---|---|---|---|---|---|
| Params(M) | 7.45 | 7.73 | 8.54 | 19.26 | 10.60 | 20.73 | 19.20 |
| FLOPs(G) | 168.29 | 379.43 | 669.65 | 113.02 | 288.04 | 749.91 | 637.45 |
| 0 | 35.05 / 0.973 | 35.47 / 0.972 | 36.46 / 0.975 | 36.58 / 0.979 | 37.09 / 0.978 | 36.67 / 0.978 | **37.63 / 0.9802** |
| 0.05 | 35.05 / 0.973 | 35.25 / 0.971 | 35.93 / 0.973 | 35.77 / 0.976 | 35.78 / 0.973 | 35.77 / 0.974 | **37.55 / 0.9800** |
| 0.1 | 35.05 / 0.973 | 34.92 / 0.970 | 35.39 / 0.972 | 35.05 / 0.973 | 34.77 / 0.969 | 35.53 / 0.973 | **37.45 / 0.9797** |
| 0.15 | 35.05 / 0.973 | 34.52 / 0.968 | 34.84 / 0.970 | 34.42 / 0.970 | 33.90 / 0.964 | 35.20 / 0.972 | **37.34 / 0.9794** |
| 0.2 | 35.05 / 0.973 | 34.08 / 0.966 | 34.29 / 0.967 | 33.84 / 0.967 | 33.15 / 0.960 | 34.81 / 0.970 | **37.21 / 0.9790** |
| 0.25 | 35.05 / 0.973 | 33.62 / 0.963 | 33.72 / 0.965 | 33.34 / 0.964 | 32.46 / 0.956 | 34.33 / 0.968 | **37.06 / 0.9786** |
| 0.3 | 35.05 / 0.973 | 33.16 / 0.960 | 33.15 / 0.962 | 32.88 / 0.961 | 31.81 / 0.951 | 33.79 / 0.965 | **36.89 / 0.9781** |
| 0.35 | 35.05 / 0.973 | 32.67 / 0.957 | 32.59 / 0.959 | 32.46 / 0.958 | 31.21 / 0.946 | 33.18 / 0.962 | **36.67 / 0.9774** |
| 0.4 | 35.05 / 0.973 | 32.16 / 0.954 | 32.03 / 0.956 | 32.08 / 0.954 | 30.66 / 0.941 | 32.53 / 0.959 | **36.40 / 0.9766** |
| 0.45 | 35.05 / 0.973 | 31.62 / 0.949 | 31.49 / 0.953 | 31.73 / 0.951 | 30.13 / 0.936 | 31.90 / 0.955 | **36.07 / 0.9755** |
| 0.5 | 35.05 / 0.973 | 31.03 / 0.944 | 30.96 / 0.949 | 31.41 / 0.948 | 29.64 / 0.930 | 31.24 / 0.951 | **35.63 / 0.9741** |

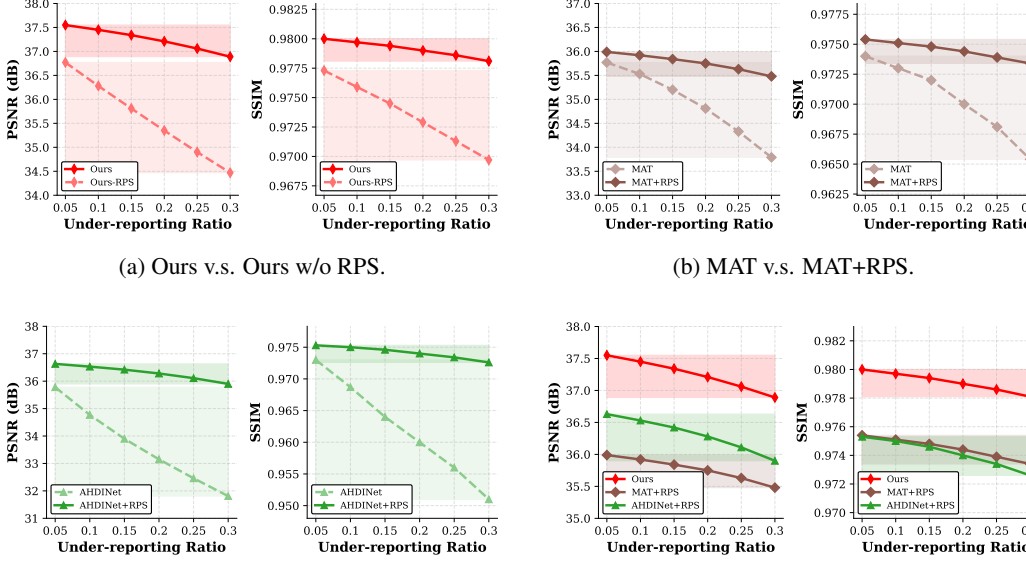

(a) Ours v.s. Ours w/o RPS.    (b) MAT v.s. MAT+RPS.

(c) AHDINet v.s. AHDINet+RPS.    (d) Ours v.s. MAT+RPS/AHDINet+RPS.

Figure 13: **Visualization of RPS verification in Table 7.** We report results of (a) Ours v.s. Ours w/o RPS, (b) MAT v.s. MAT+RPS, (c) AHDINet v.s. AHDINet+RPS, and (d) Ours v.s. MAT+RPS/AHDINet+RPS. **The following conclusions can be deduced.** (a): First, removing RPS from our framework leads to significant robustness drops. (b) and (c): Second, integrating our RPS into existing event-guided methods consistently enhances their robustness, demonstrating the plug-and-play generality of RPS. (d): Finally, our method consistently achieves the best performance and robustness, highlighting the importance of our modality-specific representation.

A.6 QUANTITATIVE AND QUALITATIVE RESULTS TO VERIFY THE EFFICIENCY OF OUR RPS

To evaluate the effectiveness of our RPS, we first validate RED and RED without RPS under different UR settings. In detail, quantitative results are listed in Table 7 and corresponding qualitative results are shown in Figure 13. Then, we implement our RPS on existing event-based methods to further verify the efficiency, including Transformer-based MAT (Xu et al., 2025) in Figure 13b and CNN-based AHDINet (Yang et al., 2025) in Figure 13c. Besides, MAT and AHDINet are individu-

Table 7: **Ablation study on RPS.** Metrics are PSNR ($\uparrow$) and SSIM ($\uparrow$).

| UR | 0.05 | 0.1 | 0.15 | 0.2 | 0.25 | 0.3 |
|---|---|---|---|---|---|---|
| DSTN (no event) | 35.05 / 0.973 | 35.05 / 0.973 | 35.05 / 0.973 | 35.05 / 0.973 | 35.05 / 0.973 | 35.05 / 0.973 |
| MAT | 35.77 / 0.974 | 35.53 / 0.973 | 35.20 / 0.972 | 34.81 / 0.970 | 34.33 / 0.9681 | 33.79 / 0.9654 |
| MAT + RPS | 35.99 / 0.9754 | 35.92 / 0.9751 | 35.84 / 0.9748 | 35.75 / 0.9744 | 35.63 / 0.9739 | 35.48 / 0.9734 |
| AHDINet | 35.78 / 0.973 | 34.77 / 0.9687 | 33.90 / 0.964 | 33.15 / 0.960 | 32.46 / 0.956 | 31.81 / 0.951 |
| AHDINet + RPS | 36.63 / 0.9753 | 36.53 / 0.9750 | 36.42 / 0.9746 | 36.28 / 0.9740 | 36.11 / 0.9734 | 35.90 / 0.9726 |
| Ours wo RPS | 36.77 / 0.9773 | 36.28 / 0.9759 | 35.81 / 0.9745 | 35.35 / 0.9729 | 34.90 / 0.9713 | 34.47 / 0.9697 |
| **Ours** | **37.55 / 0.9800** | **37.45 / 0.9797** | **37.34 / 0.9794** | **37.21 / 0.9790** | **37.06 / 0.9786** | **36.89 / 0.9781** |

ally the third-best and second-best deblurring methods when UR = 0. Finally, an overall comparison in Figure 13d is conducted on Ours, MAT with our RPS, and AHDINet with our RPS.

From Figure 13d, we draw the following conclusions. First, removing RPS from our framework leads to significant robustness drops. Second, integrating our RPS into existing event-guided methods consistently enhances their robustness, demonstrating the plug-and-play generality of RPS. Finally, beyond the contribution of RPS, our method consistently achieves the best performance and robustness across all conditions, highlighting the importance of our modality-specific representation.

### A.7 MORE QUALITATIVE RESULTS ON GOPRO, HIGHREV, AND REVD DATASETS

To provide an intuitive visualization of the deblurring performance, we present comprehensive qualitative comparisons on GoPro, HighREV, and REVD datasets. Specifically, we compare the blurry inputs and the deblurred outputs by various state-of-the-art (SOTA) methods, including our proposed RED. In addition, error maps are provided to clearly illustrate the accuracy of both spatial localization and texture restoration.

For the GoPro dataset, qualitative results are shown in Figures 14, 15, and 16. These examples cover diverse scenes such as foreground objects, background textures, and textual regions, allowing a comprehensive examination of the deblurring performance. The results demonstrate that our method consistently produces sharper edges, cleaner backgrounds, and lower error levels compared to competing methods, highlighting its superior capability in precise detail reconstruction.

For the unseen real-world scenarios in the HighREV and REVD datasets, qualitative comparisons are presented in Figures 17 and 18. These datasets contain challenging cases with degraded or noisy event data, where robustness becomes crucial. Interestingly, we observe that for certain existing methods, the deblurred outputs can be even worse than the original blurry inputs, as reflected by lower visual quality and higher error levels.

As shown in Figure 17, EFNet and MAT almost fail to perform meaningful deblurring, producing results that are close to the original blurry inputs. In contrast, STCNet, AHDINet, and TRMD are able to restore coarse object contours, but their outputs still suffer from structural distortions, such as misaligned edges and inconsistent textures. For the REVD dataset in Figure 18, we observe different failure patterns: EFNet and MAT exhibit over-sharpening, generating noticeable artifacts and non-existent edges. It is worth noted that STCNet produces results that are visually even worse than the blurry inputs, likely because its design of three-frame rolling input introduces a higher degree of unseen and challenging events, thereby degrading the output quality. TRMD performs similarly to the blurry images, with only subtle pixel-level improvements that are barely visible to the naked eye, primarily around the right edge of the wall in Figure 18. Among all methods, AHDINet and Ours achieve relatively better deblurring performance, with Ours demonstrating sharper edges and more faithful details, such as clearer hair strands and finer clothing textures.

The above observations suggest that the event information, without modality-specific representation, may fail to provide positive guidance and may instead introduce artifacts. With our RPS and modality-specific MRM, our approach maintains superior deblurring performance, producing clearer edges and more faithful texture details across diverse and challenging conditions.

Table 8: **Performance under Under-Reporting Degradation.** Metrics are PSNR (↑) and SSIM (↑).

| UR | DSTN | EFNet | STCNet | TRMD | AHDINet | MAT | Ours |
|---|---|---|---|---|---|---|---|
| Params(M) | 7.45 | 7.73 | 8.54 | 19.26 | 10.60 | 20.73 | 19.20 |
| FLOPs(G) | 168.29 | 379.43 | 669.65 | 113.02 | 288.04 | 749.91 | 637.45 |
| 0.00 | 35.05/0.973 | 35.47/0.972 | 36.46/0.975 | 36.58/0.979 | 37.09/0.978 | 36.67/0.978 | **37.63/0.9802** |
| 0.05 | 35.05/0.973 | 35.25/0.971 | 35.93/0.973 | 35.77/0.976 | 35.78/0.973 | 35.77/0.974 | **37.55/0.9800** |
| 0.10 | 35.05/0.973 | *34.92/0.970* | 35.39/0.972 | 35.05/0.973 | *34.77/0.9687* | 35.53/0.973 | **37.45/0.9797** |
| 0.15 | 35.05/0.973 | – | *34.84/0.970* | *34.42/0.970* | – | 35.20/0.972 | **37.34/0.9794** |
| 0.20 | 35.05/0.973 | – | – | – | – | *34.81/0.970* | **37.21/0.9790** |
| 0.25 | 35.05/0.973 | – | – | – | – | – | **37.06/0.9786** |
| 0.30 | 35.05/0.973 | – | – | – | – | – | **36.89/0.9781** |
| 0.35 | 35.05/0.973 | – | – | – | – | – | **36.67/0.9774** |
| 0.40 | 35.05/0.973 | – | – | – | – | – | **36.40/0.9766** |
| 0.45 | 35.05/0.973 | – | – | – | – | – | **36.07/0.9755** |
| 0.50 | 35.05/0.973 | – | – | – | – | – | **35.63/0.9741** |
| 0.55 | 35.05/0.973 | – | – | – | – | – | **35.06/0.9721** |
| 0.60 | **35.05/0.973** | – | – | – | – | – | *34.34/0.9693* |

Table 9: **Performance under Temporal Jitter Degradation.** Metrics are PSNR (↑) and SSIM (↑).

| Jitter | DSTN | EFNet | STCNet | TRMD | MAT | AHDINet | Ours |
|---|---|---|---|---|---|---|---|
| Params(M) | 7.45 | 7.73 | 8.54 | 19.26 | 10.60 | 20.73 | 19.20 |
| FLOPs(G) | 168.29 | 379.43 | 669.65 | 113.02 | 288.04 | 749.91 | 637.45 |
| 0.1 | 35.05/0.973 | 35.44/0.9720 | 35.71/0.972 | 36.46/0.978 | 35.95/0.9748 | 36.54/0.975 | **37.50/0.9798** |
| 0.2 | 35.05/0.973 | 35.38/0.9718 | *34.43/0.966* | 36.17/0.977 | 35.91/0.9747 | 36.23/0.974 | **37.18/0.9790** |
| 0.3 | 35.05/0.973 | 35.29/0.9716 | – | 35.80/0.976 | 35.85/0.9745 | 35.95/0.973 | **36.84/0.9780** |
| 0.4 | 35.05/0.973 | 35.15/0.9711 | – | 35.39/0.974 | 35.76/0.9741 | 35.56/0.971 | **36.39/0.9766** |
| 0.5 | 35.05/0.973 | *34.95/0.9702* | – | 34.82/0.971 | 35.62/0.9746 | 35.16/0.969 | **35.92/0.9749** |
| 0.6 | 35.05/0.973 | – | – | – | 35.42/0.9729 | *34.81/0.967* | **35.51/0.9729** |
| 0.7 | **35.05/0.973** | – | – | – | *35.03/0.9710* | – | *35.01/0.9708* |

## A.8   REBUTTAL MATERIALS: GENERALIZATION OF RPS TO OTHER SENSOR DEGRADATIONS

We compare representative event-based methods and our RED under three degradations: under-reporting, temporal jitter, and background noise. The image-only performance serves as a reference baseline.

From Table 8, baselines degrade quickly as the ratio increases, some reaching the image-only baseline around UR ≈ 0.1–0.15. RED maintains the most stable behavior up to UR = 0.6.

From Table 9, mild jitter (0.1–0.2) affects only a few methods, while stronger jitter leads to rapid performance drop for most baselines. RED remains consistently superior across all jitter levels.

From Table 10, most baselines collapse as noise approaches 0.1, while RED maintains the smoothest and highest overall performance.

Across under-reporting, temporal jitter, and noise, several baselines remain stable under mild perturbations, but most suffer rapid degradation as the perturbation increases. **RED maintains stable and superior performance across the entire perturbation range**, consistently outperforming all baselines and remaining above the image-only baseline even under heavy degradations. These results demonstrate that RPS (input-level perturbation) and MRM (representation-level decoupling) provide structural robustness that generalizes naturally beyond under-reporting.

Table 10: **Performance under Background Noise Degradation.** Metrics are PSNR (↑) and SSIM (↑).

| Noise | DSTN | EFNet | STCNet | TRMD | MAT | AHDINet | Ours |
|---|---|---|---|---|---|---|---|
| Params(M) | 7.45 | 7.73 | 8.54 | 19.26 | 10.60 | 20.73 | 19.20 |
| FLOPs(G) | 168.29 | 379.43 | 669.65 | 113.02 | 288.04 | 749.91 | 637.45 |
| 0.00 | 35.05/0.973 | 35.47/0.972 | 36.46/0.975 | 36.58/0.979 | 37.09/0.978 | 37.09/0.978 | **37.61/0.9801** |
| 0.01 | 35.05/0.973 | 35.12/0.970 | 36.36/0.975 | 36.58/0.9786 | 35.62/0.974 | 37.06/0.9776 | **37.58/0.9799** |
| 0.02 | 35.05/0.973 | *32.64/0.957* | 36.09/0.973 | 36.57/0.9785 | *34.97/0.970* | 36.86/0.9765 | **37.54/0.9796** |
| 0.03 | 35.05/0.973 | – | 35.80/0.972 | 36.55/0.9783 | – | 36.47/0.974 | **37.45/0.9791** |
| 0.04 | 35.05/0.973 | – | 35.52/0.970 | 36.53/0.9781 | – | 35.97/0.9705 | **37.31/0.9783** |
| 0.05 | 35.05/0.973 | – | 35.24/0.968 | 36.48/0.9776 | – | 35.45/0.966 | **37.12/0.9771** |
| 0.06 | 35.05/0.973 | – | *34.98/0.965* | 36.41/0.977 | – | *34.96/0.962* | **36.85/0.9756** |
| 0.07 | 35.05/0.973 | – | – | 36.30/0.976 | – | – | **36.52/0.9737** |
| 0.08 | 35.05/0.973 | – | – | 36.14/0.973 | – | – | **36.31/0.9725** |
| 0.09 | 35.05/0.973 | – | – | 35.91/0.970 | – | – | **36.13/0.9716** |
| 0.10 | 35.05/0.973 | – | – | 35.62/0.966 | – | – | **35.70/0.9693** |
| 0.11 | 35.05/0.973 | – | – | 35.28/0.962 | – | – | **35.27/0.9669** |
| 0.12 | **35.05/0.973** | – | – | *34.95/0.957* | – | – | *34.84/0.9645* |

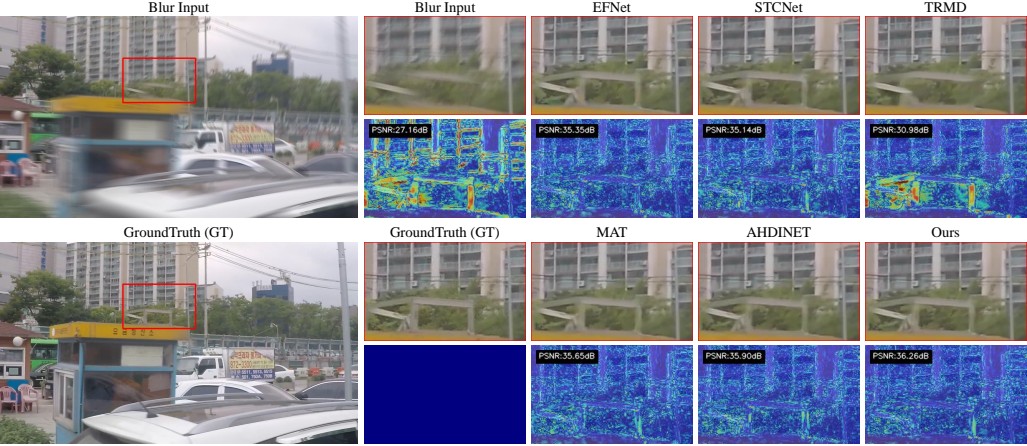

Figure 14: **Visualization of an example of background texture in GoPro dataset.** Besides, error maps are drawn to illustrate a comprehensive comparison of both localization and structure context.

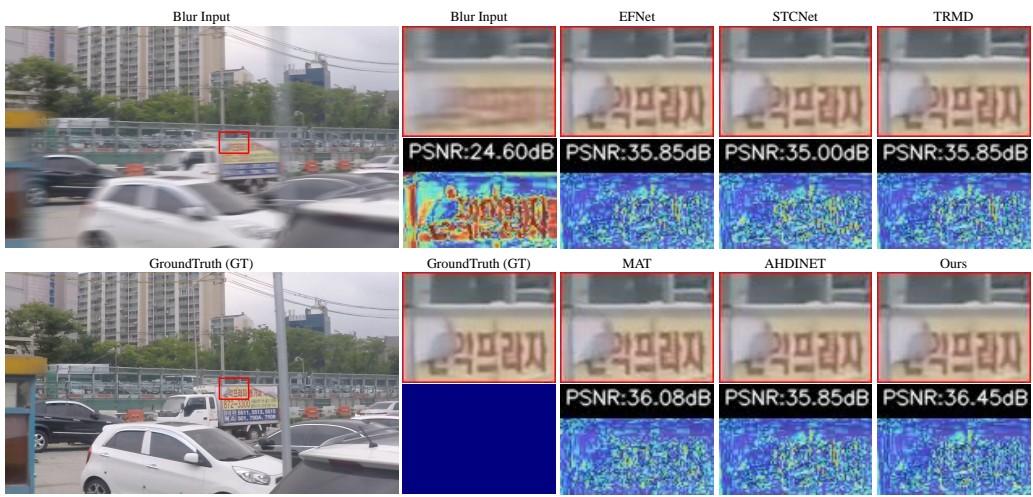

Figure 15: **Visualization of an example of textual region in GoPro dataset.** Besides, error maps are drawn to illustrate a comprehensive comparison of both localization and structure context.

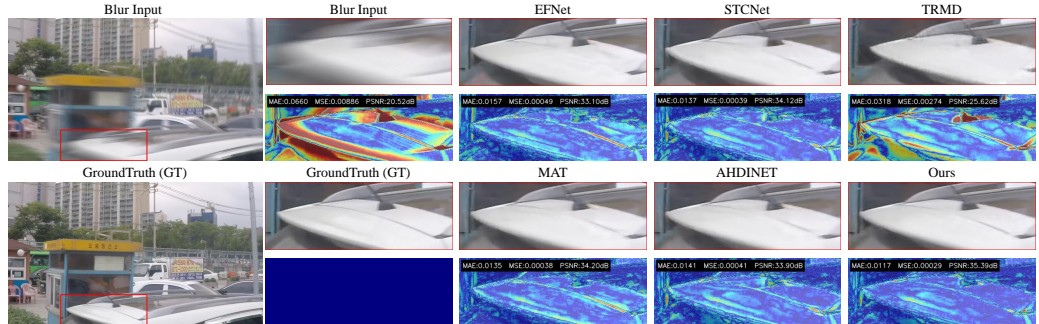

Figure 16: **Visualization of an example of foreground object in GoPro dataset.** Besides, error maps are drawn to illustrate a comprehensive comparison of both localization and structure context.

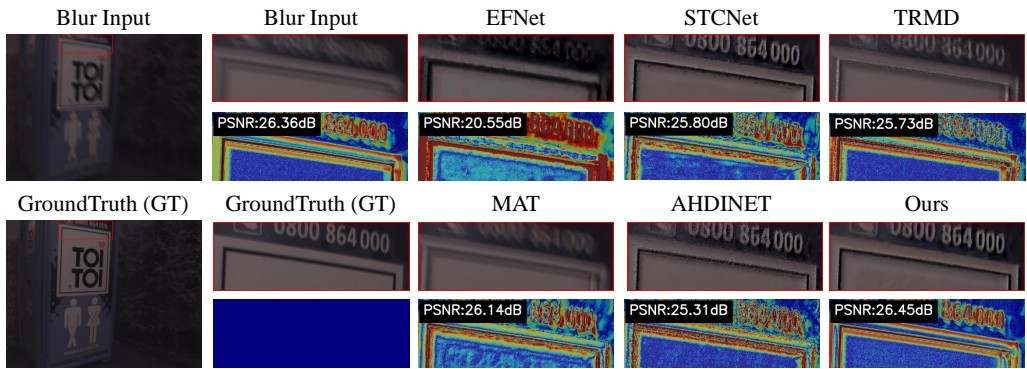

Figure 17: **Visualization in HighREV dataset.** Besides, error maps are drawn to illustrate a comprehensive comparison of both localization and structure context. **We can infer that:** EFNet and MAT almost fail to produces results that are close to the original blurry inputs. In contrast, STCNet, AHDINet, and TRMD are able to restore coarse object contours, but suffer from misaligned edges and inconsistent textures. Relatively, Ours achieve a clearer and sharp result.

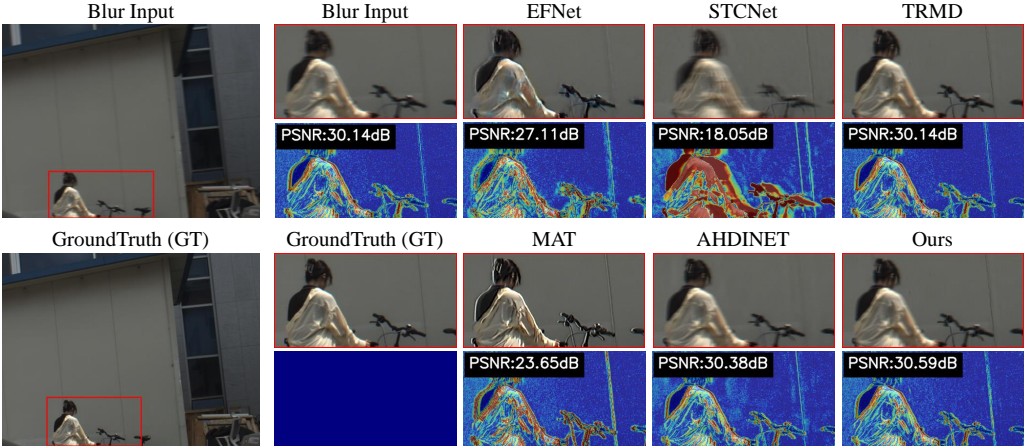

Figure 18: **Visualization in REVD dataset.** Besides, error maps are drawn to illustrate a comprehensive comparison of both localization and structure context. **We observe that:** EFNet and MAT tend to over-sharpen. STCNet produces results that are visually even worse than the blurry inputs since the three-frame rolling strategy causes a higher degree of events. TRMD performs similarly to the blurry images, with only subtle pixel-level improvements, primarily around the right edge of the wall. Among all methods, AHDINet and Ours achieve relatively better deblurring performance, with Ours demonstrating clearer hair strands and finer clothing textures.

