# OpenReview forum: "RED: Robust Event-Guided Motion Deblurring with Modality-Specific Disentangled Representation"
_ICLR.cc/2026/Conference — Submitted to ICLR 2026_

### Official Review · Reviewer_bp3s · 2025-10-31

**Soundness:** 2
**Presentation:** 3
**Contribution:** 2
**Rating:** 6
**Confidence:** 3

**Summary:**

Considering the trade-off between noisy fake event and under-reporting of events, this paper proposes a robust event-guided deblurring network with modality-specific disentangled representation. It improves the network robustness to different DVS thresholds (diverse under-reporting patterns) with a perturbation strategy. Besides, it presents the blurry image and partially disrupted event data with modality-specific representations. It also introduces two interactive modules to enhance motion-sensitive areas in blurry images.

**Strengths:**

1, It proposes a robust event-guided motion deblurring framework, which is trained with various DVS thresholds.

2, It disentangles modality-specific representation for image and event data. These two modalities interacts and benefits each other with two modules: one for improving spatial details in images and the other one for enhancing semantic information in event encoding.

**Weaknesses:**

Overall, this paper is well-written and its designs are validated with various experiments. However, I have some concerns in terms of the core contributions.


1, The perturbation strategy looks like a data augmentation trick.

2, The novelty of disentangleed image and event representations is arguable. Previous methods already follow the three steps of image feature extraction, event feature extraction and cross-module aggregation.

3, The cross modality attention is the common cross attention. In addition, it is strange to compute the query and key from one modality, and obtain the value from another modality. Generally, key and value should come from one modality.

4, The abstract could be improved with more details on the modality-specific representations.

**Questions:**

Please see weakness.

---

> ### Author Response · Authors · 2025-11-24
>
> # Q1. RPS is fundamentally different from conventional data augmentation
>
> Thank you for the reviewer’s insightful comments. In response to the concern that “RPS looks like a data augmentation trick,” we clarify the nature and role of our Robustness-Oriented Perturbation Strategy (RPS) from three complementary perspectives: its theoretical motivation, its consistent cross-architecture empirical effects, and its synergy with our model design.
>
> ## 1. Theoretical motivation
>
> The design of RPS is grounded in the analysis presented in Lines 183–231 of our paper, where we study how different DVS threshold settings naturally lead to under-reporting of events. Real event cameras inherently produce varying levels of event missing due to the physical triggering mechanism, and such sensor-driven dropout patterns cannot be reproduced by typical image-level augmentations such as random noise injection, masking, or dropout.
>
> Based on this mechanism, RPS is introduced to explicitly simulate threshold-induced event dropout patterns during training, allowing the model to learn event representations that are invariant to threshold variations. RPS is architecture-agnostic, parameter-free, and applied only during training. Therefore, its underlying motivation, operational mechanism, and intended effect are fundamentally different from traditional data augmentation.
>
> ## 2. Cross-architecture empirical validation shows RPS goes beyond “augmentation”
>
> To ensure that the benefits of RPS do not originate from changes in training duration or data exposure, we kept all training configurations identical across all methods. This includes the commonly used training schedule of 2000 iterations × 200 epochs, the 256×256 patch size, and all other settings.
>
> We applied RPS to two structurally different event-guided deblurring models, the CNN-based AHDINet and the Transformer-based MAT, while keeping all training settings and data identical. The results clearly show that RPS consistently improves both architectures across all under-reporting ratios, with even larger improvements under higher dropout conditions. This systematic and architecture-independent improvement cannot be explained by generic augmentation effects and instead aligns with the intended modeling of real sensor-induced event dropout.
>
> The full quantitative results (PSNR / SSIM) under different under-reporting ratios (UR) are shown below:
>
> | **UR** | **DSTN (no event)** | **MAT**        | **MAT+RPS**     | **AHDINet**    | **AHDINet+RPS** | **Ours w/o RPS** | **Ours**            |
> |--------|-----------------------|----------------|------------------|-----------------|------------------|-------------------|----------------------|
> | 0.05   | 35.05/0.973           | 35.77/0.974    | 35.99/0.9754     | 35.78/0.973     | 36.63/0.9753     | 36.77/0.9773       | **37.55/0.9800**     |
> | 0.10   | 35.05/0.973           | 35.53/0.973    | 35.92/0.9751     | 34.77/0.9687    | 36.53/0.9750     | 36.28/0.9759       | **37.45/0.9797**     |
> | 0.15   | 35.05/0.973           | 35.20/0.972    | 35.84/0.9748     | 33.90/0.964     | 36.42/0.9746     | 35.81/0.9745       | **37.34/0.9794**     |
> | 0.20   | 35.05/0.973           | 34.81/0.970    | 35.75/0.9744     | 33.15/0.960     | 36.28/0.9740     | 35.35/0.9729       | **37.21/0.9790**     |
> | 0.25   | 35.05/0.973           | 34.33/0.9681   | 35.63/0.9739     | 32.46/0.956     | 36.11/0.9734     | 34.90/0.9713       | **37.06/0.9786**     |
> | 0.30   | 35.05/0.973           | 33.79/0.9654   | 35.48/0.9734     | 31.81/0.951     | 35.90/0.9726     | 34.47/0.9697       | **36.89/0.9781**     |
>
>
> These results demonstrate that RPS provides stable and significant improvements across different network architectures. Such architecture-agnostic consistent gains indicate that RPS is not general data augmentation but an effective simulation of the physical event dropout mechanism.
>
> ## 3. Why our method remains the best-performing even after adding RPS to MAT and AHDINet
>
> Although RPS improves MAT and AHDINet, our method still achieves the best performance. This indicates that merely exposing the model to disrupted events is insufficient for achieving optimal deblurring under high under-reporting levels. The crucial factor is whether the model can extract stable and meaningful motion features from corrupted event sequences.
>
> Existing methods often use the same backbone for both the RGB and event modalities, but disrupted events differ drastically from RGB images in distribution and reliability. Such architectures lack modality-specific feature extraction capability, which limits their ability to make effective use of corrupted events even when trained with RPS.

---

> ### Author Response · Authors · 2025-11-24
>
> # Q2  Explaination of disentangled image and event representations
>
> Thank you for the insightful comments. While the overall three-stage pipeline of *image feature extraction → event feature extraction → cross-modal fusion* appears similar to prior frameworks, our contribution does not lie in rearranging this pipeline. Instead, our innovation centers on **how representations are separated, how each modality is modeled, and how cross-modal compensation is performed under disrupted-event conditions**, which resolves a core limitation that previous methods do not address.
>
> Current event-guided deblurring methods indeed follow the three-stage pipeline. However, most prior approaches do **not** structurally differentiate image and event representations. They typically adopt identical or nearly identical backbones for both modalities. As a result, the extractors are not adapted to the fundamentally different statistical properties of images and events. When event data suffer from under-reporting or structural degradation, the event-branch representation becomes unstable, and the static semantic features from images tend to mix with the dynamic motion cues from events. This mixed representation may still work when events are complete, but it significantly weakens motion estimation once events are disrupted.
>
> To address this intrinsic limitation, we revisit the event-guided deblurring task and propose a **modality-specific first, then modality-coupled** representation strategy. We design modality-specific extractors tailored for the two modalities. The image branch first builds stable semantic structures, while the event branch focuses on temporal changes and sparse motion signatures. After each modality forms its own private representation, a targeted fusion stage is applied to prevent static semantics and dynamic motion cues from interfering with one another. This ensures that the fused representation remains stable even when the event modality becomes degraded.
>
> A second key contribution is our semantic-guided cross-modal compensation mechanism. The cross-attention module in our system is not the conventional symmetric design in which both modalities are used to generate the query, key, and value. Instead, we explicitly design the cross-attention based on the observation that the event modality loses semantic understanding when dropout occurs. Our goal is to use the rich semantic structure from the image branch to compensate for degraded event representations. Therefore, both the query and the key are obtained from image features to construct a semantic attention map:
>
> $$
> \mathbf{A} _{I \rightarrow E} = \mathrm{Softmax}(\mathbf{Q} _I \mathbf{K} _I^\top),
> $$
>
> and this semantic prior is projected onto the event representation:
>
> $$
> \mathbf{\tilde{F}} _E = \mathbf{V} _E \odot \mathbf{A} _{I \rightarrow E}
> $$
>
> This design is fundamentally different from traditional symmetric cross-attention. Our mechanism implements **semantic transfer** from image to event features. The image modality serves as a semantic anchor that stabilizes event representations degraded by under-reporting. Our experiments show that this semantic reinforcement is essential for robust reconstruction under high disrupted ratios.
>
> To validate this behavior, we visualize the activation maps of both branches before and after MSEM/ESEM in **Figure 3 and Figure 11**, including: (1) image-encoder features before and after MSEM, (2) image-branch decoder features, (3) event-encoder features before and after ESEM, and (4) event-branch decoder features. The visualizations reveal the following patterns:
> - **Before MSEM**, image activations exhibit broad semantic layouts, while event activations focus on high-motion regions.
> - **After MSEM**, the image branch preserves its semantic structure, but exhibits enhanced responses exactly at event-indicated motion areas.
> - **After ESEM**, the event branch receives semantic reinforcement from the image branch, producing motion representations with more complete contours and improved structural continuity.
> - **In the decoder**, the image modality consistently governs global structure and coherence, while the event modality strengthens local detail recovery and edge transitions.
>
> These observations demonstrate that the primary functional separation between semantic and motion cues remains intact throughout the pipeline, and that MSEM/ESEM operate as intended: **providing directed cross-modal compensation without collapsing the disentangled representation space**.

---

> ### Author Response · Authors · 2025-11-24
>
> # Q3  Value and Key design in our cross-modality Attention
>
> Thank you for detailed observation. The cross-attention module used in our method is not the conventional form of “symmetrically generating query–key–value from both modalities without distinction.” Instead, it is deliberately designed around the challenges of image deblurring under disrupted events: on one hand, the event modality suffers from severely weakened semantic capacity under dropout; on the other hand, sparse and degraded events still contain crucial dynamic structural priors that must be effectively extracted to guide the image deblurring process.
>
> First, the event modality lacks global semantic structure under high under-reporting ratios, whereas the image modality still provides stable and complete semantic priors. Therefore, when transferring semantic cues from images to events, both the query and key are taken from the image modality to construct a reliable semantic attention map:
> $$
> \mathbf{A} _{I \rightarrow E} = \mathrm{Softmax}(\mathbf{Q} _{I}\mathbf{K} _{I}^{\top})
> $$
>
> and this semantic prior is then projected onto the event modality via:
>
> $$
> \widetilde{\mathbf{F}} _{E} = \mathbf{V} _{E} \odot \mathbf{A} _{I \rightarrow E}
> $$
>
> This design is not a conventional cross-attention mechanism but a form of semantic-transfer reinforcement, whose core purpose is to supplement the event representations with global semantic structure that is severely missing under dropout. Our experiments show that such one-way semantic compensation is crucial for robust reconstruction under high DR conditions.
>
> Second, although the event modality is weak in semantics, it is highly sensitive to temporal variations and motion boundaries, providing dynamic structural information that is often missing in the image branch. Therefore, when enhancing the image modality with motion cues, we adopt a cross-attention in the opposite direction, where the query and key are derived from the event modality:
>
> $$
> \mathbf{A} _{E \rightarrow I} = \mathrm{Softmax}(\mathbf{Q} _{E}\mathbf{K} _{E}^{\top})
> $$
>
> and the value comes from the image modality:
>
> $$
> \widetilde{\mathbf{F}} _{I} = \mathbf{V} _{I} \odot \mathbf{A} _{E \rightarrow I}
> $$
>
> This allows the event modality, with its high temporal resolution, to guide the image reconstruction and restore sharper structures and edges, rather than relying on a conventional symmetric cross-attention. In this way, the motion-sensitive characteristics of events are fully exploited to provide strong structural priors when the image modality is degraded.
>
> In summary, the two cross-attention directions used in our method are not a simple reuse of generic cross-attention, but two task-specific and asymmetrically designed prior-transfer mechanisms that address the dual challenges of semantic insufficiency in events and structural insufficiency in images. Our choice of different query–key–value sources in the two directions is not an incorrect use of cross-attention, but a principled modeling decision based on the complementary strengths of the two modalities. This design enables semantic compensation and structural enhancement to function simultaneously, leading to significantly improved deblurring performance under disrupted event scenarios.
>
> -----------------------------------------------------------------------------------------
> # Q4  Abstract Improvement
> Thank you for the constructive suggestion. We revise the abstract according to the comments.

---

### Official Review · Reviewer_gaMS · 2025-10-31

**Soundness:** 4
**Presentation:** 3
**Contribution:** 3
**Rating:** 6
**Confidence:** 4

**Summary:**

This paper introduces a robustness-oriented perturbation strategy (RPS) for event-based vision networks, aiming to improve their resilience under event under-reporting or dropout conditions. RPS dynamically simulates changes in the DVS output threshold to enable the network to adapt to diverse dropout patterns.

To further address feature unreliability caused by indiscriminate fusion or the absence of weak events, the authors propose a modality-specific representation mechanism (MRM) that disentangles semantic, motion, and cross-modality features. Two specialized submodules enable “coadjutant interactions”:

1. Motion Saliency Enhancer Module (MSEM) – strengthens motion-related spatial details often lost in blur.

2. Event Semantic Engraver Module (ESEM) – transforms semantic cues from images into deep event embeddings, mitigating semantic degradation caused by sparse events.

The paper also provides a probabilistic analysis of noise aggregation, modeling photon arrival with a Poisson process and circuit noise as Gaussian. Through comprehensive ablation and cross-model studies on several benchmarks, the proposed approach shows clear improvements in PSNR and SSIM, confirming both robustness and generalization.

**Strengths:**

The RPS module introduces a practical, model-agnostic way to simulate and handle event dropouts, effectively bridging the gap between synthetic and real-world noise conditions.

RPS can be easily integrated into other frameworks (e.g., MAT, AHDINet) and consistently improves their performance without architectural redesign.

The disentanglement design via MRM, combined with MSEM and ESEM, leads to well-structured multimodal representation and enhanced reconstruction fidelity.

Ablation and cross-model experiments rigorously validate the contribution of each component, demonstrating robustness across under-reporting ratios.

The model achieves superior PSNR/SSIM, especially under high dropout rates (>0.2).

**Weaknesses:**

Computational Cost Not Reported – The extra cost introduced by RPS (e.g., FLOPs, inference delay) is not discussed, which is crucial for real-time applications.

Limited Perturbation Scope – The experiments focus solely on under-reporting; robustness against other real-world degradations (e.g., motion blur, temporal jitter, or sensor noise) is untested.

Weak Theoretical Analysis – The mechanism by which RPS and MRM improve robustness lacks a formal mathematical explanation or interpretability study.

Partial Feature Disentanglement – Although semantic and motion features are decoupled in design, MSEM and ESEM remain interdependent through attention coupling, suggesting only dominant (not complete) disentanglement.

**Questions:**

How much computational overhead (in FLOPs or runtime) does RPS introduce during training and inference?

Could the RPS concept be extended to handle other sensor degradations, such as background noise or temporal jitter?

How independent are the features extracted by MSEM and ESEM? Have you visualized their activation maps to confirm disentanglement?

Can the authors provide more intuition or theoretical justification on how dynamic threshold perturbation leads to feature stability?

Would combining RPS with other noise-injection or adversarial training methods yield further robustness gains?

---

> ### Author Response · Authors · 2025-11-24
>
> # W1/Q1 — Computational Cost Introduced by RPS: Analysis and Empirical Validation
>
> Thank you for raising this important concern regarding the computational overhead of RPS. We answer this question from both theoretical analysis and empirical measurement.
>
> RPS is **architecture-agnostic** and **parameter-free**, designed to simulate diverse under-reporting patterns caused by different DVS thresholds. Following the introduction in Lines 182–239 of the paper, the theoretical computational cost of RPS is:
>
> $$
> FLOPs = TCHW + 3TCHW \cdot DR = TCHW (1 + 3DR)
> $$
>
> where $T$ is the number of event frames, $(C, H, W)$ are the channel and spatial dimensions, and $DR$ denotes the disrupted ratio.
>
> To further quantify the overhead, we test the runtime of RPS on an event tensor of size $(B, T, C, H, W) = (1, 6, 1, 360, 640)$. The averaged results over 100 runs are listed below:
>
> --------------------------------------------------------
> **Table 1: Runtime and FLOPs of RPS Under Different Disrupted Ratios**
>
> | DR   | Avg Time (ms) | Std (ms) | FLOPs (M) |
> |------|---------------|----------|-----------|
> | 0.0  | 0.666         | 0.005    | 1.56      |
> | 0.1  | 0.710         | 0.004    | 2.02      |
> | 0.2  | 0.710         | 0.003    | 2.49      |
> --------------------------------------------------------
>
> Even at the maximum disrupted ratio used during training (DR = 0.2), RPS introduces only **2.49M additional FLOPs** and roughly **0.71 ms** of runtime overhead when applied to the above tensor.
>
> Overall, both theoretical derivation and empirical validation confirm that RPS brings only **affordable extra computation**, making it suitable even for practical applications.
>
> --------------------------------------------------------
>
> # W4/Q3 — Independence of MSEM and ESEM Features and Visualization-Based Evidence
>
> Thank you for your comments, as well as for suggesting the use of activation visualizations. We address this from both the mechanism design perspective and empirical visualization results.
>
> Our overall design philosophy is to **first establish a clear semantic–motion disentanglement** in MRM, and then allow **two complementary targeted cross-modal compensation** where each modality selectively contributes only the most task-relevant cues to the other. In this formulation, MSEM and ESEM are not intended to remap or re-mix the already disentangled representations; instead, they inject complementary information in a controlled manner **without altering the primary functional roles of the semantic and motion branches**.
>
> Specifically, MSEM uses the naturally high-response motion regions from events to guide the image branch. This enhances attention to motion-sensitive areas while preserving the global semantic structure extracted from images. Conversely, ESEM draws high-level semantics from the image branch to compensate for the under-reporting or sparsity of events, helping the event encoder build more complete and coherent motion contours. Because these interactions are direction-specific and compensation-oriented: semantic content remains governed by the image pathway, and motion cues remain governed by the event pathway. The cross-modal attention does not override or dilute these roles.
>
> To validate this behavior, we visualize the activation maps of both branches before and after MSEM/ESEM, including: (1) image-encoder features before and after MSEM, (2) image-branch decoder features, (3) event-encoder features before and after ESEM, and (4) event-branch decoder features. The visualizations reveal the following patterns:
> - **Before MSEM**, image activations exhibit broad semantic layouts, while event activations focus on high-motion regions.
> - **After MSEM**, the image branch preserves its semantic structure, but exhibits enhanced responses exactly at event-indicated motion areas.
> - **After ESEM**, the event branch receives semantic reinforcement from the image branch, producing motion representations with more complete contours and improved structural continuity.
> - **In the decoder**, the image modality consistently governs global structure and coherence, while the event modality strengthens local detail recovery and edge transitions.
>
> These observations demonstrate that the primary functional separation between semantic and motion cues remains intact throughout the pipeline, and that MSEM/ESEM operate as intended: **providing directed cross-modal compensation without collapsing the disentangled representation space**.
>
> Based on this feedback, we have added the corresponding visualizations and a clearer explanation to the revised manuscript (**Figure 3 and Figure 11**). These additions further support that MSEM and ESEM do not compromise the disentanglement established by MRM, but instead reinforce each modality’s strengths in a controlled and interpretable manner.

---

> ### Author Response · Authors · 2025-11-24
>
> # W2/Q2(Part 1) — Generalization of RPS to Other Sensor Degradations and Robustness Analysis
>
> Thank you for this important question. Our response is organized into three parts: (1) degradations already present in the evaluated datasets, (2) why threshold-induced under-reporting is a fundamental DVS mechanism, (3) robustness under multiple perturbations.
>
> ---
>
> ## ① Degradations Already Present in the Datasets
>
> Although the question focuses on under-reporting, the datasets used in our evaluation naturally include broader sensor degradations,
>
> - **GoPro**: A widely used benchmark for dynamic-scene deblurring. Blurry images are generated by averaging consecutive high-speed frames captured at 240 fps, producing realistic motion blur in natural scenes.
> - **HighREV**:  A high-resolution dataset containing real event streams paired with color video. It captures challenging real-world motion and illumination variations to support event-based interpolation and reconstruction tasks.
> - **REVD**: A real-world dataset for event-based video deblurring that provides paired event data and motion-blurred frames, collected under diverse real-scene dynamics to evaluate robust deblurring under practical conditions.
>
> Together, these datasets naturally cover dynamic-scene motion blur (GoPro), real event-RGB modality characteristics under fast motion (HighREV), and real-world motion-blur degradation paired with event streams (REVD), providing a broad evaluation space beyond under-reporting alone.
>
> ---
>
> ## ② Why Under-Reporting Is a Fundamental Degradation for Event Sensors
>
> From the derivation in Lines 182–231, the contrast-trigger mechanism satisfies:
>
> $$ \theta \uparrow \Rightarrow \text{noise} \downarrow \; \text{under-reporting} \uparrow. $$
>
> This means noise and under-reporting are jointly controlled by the same threshold.
> Results in Fig. 1(c)(d) further show that noise ≈ 0.1 severely destabilizes existing event-based methods, while under-reported events remain more reliable and still encode high-confidence motion cues.
> Therefore, under-reporting is not an isolated degradation but a core behavior of DVS sensors.
>
>
> ## ③ Robustness Under Multiple Sensor Degradations
>
> We compare representative event-based methods and our RED under three degradations: under-reporting, temporal jitter, and background noise.
> The detailed results are shown in Tables 8, 9, and 10, where the image-only performance serves as a reference baseline. Below are comparisons of PSNR.
>
> ---
>
> ### **(a) Under-Reporting**
>
> From Table 1, baselines degrade quickly as the ratio increases, some reaching the image-only baseline around UR ≈ 0.1–0.15. RED maintains the most stable behavior up to UR = 0.6.
>
> **Table 1. Performance under Under-Reporting (PSNR)**
>
> ----------------------------------------
> | UR   | DSTN(no event) | EFNet | STCNet | TRMD | AHDINet | MAT  | Ours |
> |:---:|:-----------------:|:-----:|:------:|:----:|:-------:|:----:|:----:|
> | 0.00  | 35.05   | 35.47   | 36.46   | 36.58   | 37.09    | 36.67   | **37.63**   |
> | 0.05  | 35.05   | 35.25   | 35.93   | 35.77   | 35.78    | 35.77   | **37.55**   |
> | 0.10  | 35.05   | **34.92** | 35.39   | 35.05   | **34.77** | 35.53   | **37.45**   |
> | 0.15  | 35.05   | -       | **34.84** | **34.42** | -        | 35.20   | **37.34**   |
> | 0.20  | 35.05   | -       | -       | -       | -        | **34.81** | **37.21**   |
> | 0.25  | 35.05   | -       | -       | -       | -        | -       | **37.06**   |
> | 0.30  | 35.05   | -       | -       | -       | -        | -       | **36.89**   |
> | 0.35  | 35.05   | -       | -       | -       | -        | -       | **36.67**   |
> | 0.40  | 35.05   | -       | -       | -       | -        | -       | **36.40**   |
> | 0.45  | 35.05   | -       | -       | -       | -        | -       | **36.07**   |
> | 0.50  | 35.05   | -       | -       | -       | -        | -       | **35.63**   |
> | 0.55  | 35.05   | -       | -       | -       | -        | -       | **35.06**   |
> | 0.60  | *35.05* | -       | -       | -       | -        | -       | **34.34**   |

---

> ### Author Response · Authors · 2025-11-24
>
> # W2/Q2(Part 2) — Generalization of RPS to Other Sensor Degradations and Robustness Analysis
>
> ## **(b) Temporal Jitter**
>
> From Table 2, mild jitter (0.1–0.2) affects only a few methods, while stronger jitter leads to rapid performance drop for most baselines. RED remains consistently superior across all jitter levels.
>
> **Table 2. Performance under Temporal Jitter (PSNR)**
>
> ----------------------------------------
> | Jitter | DSTN(no event)   | EFNet   | STCNet | TRMD  | MAT    | AHDINet | Ours        |
> |:---:|:-----------------:|:-----:|:------:|:----:|:-------:|:----:|:----:|
> | 0.1    | 35.05  | 35.44    | 35.71   | 36.46  | 35.95   | 36.54    | **37.50**   |
> | 0.2    | 35.05  | 35.38    | **34.43** | 36.17  | 35.91   | 36.23    | **37.18**   |
> | 0.3    | 35.05  | 35.29    | -       | 35.80  | 35.85   | 35.95    | **36.84**   |
> | 0.4    | 35.05  | 35.15    | -       | 35.39  | 35.76   | 35.56    | **36.39**   |
> | 0.5    | 35.05  | **34.95** | -       | 34.82  | 35.62   | 35.16    | **35.92**   |
> | 0.6    | 35.05  | -        | -       | -      | 35.42   | **34.81** | **35.51**   |
> | 0.7    | *35.05*| -        | -       | -      | **35.03** | -        | **35.01**   |
> ----------------------------------------
>
> ---
>
> ## **(c) Background Noise**
>
> From Table 3, most baselines collapse as noise approaches 0.1, while RED maintains the smoothest and highest overall performance.
>
> **Table 3. Performance under Background Noise (PSNR)**
>
> ----------------------------------------
> | Noise | DSTN(no event)   | EFNet   | STCNet | TRMD  | MAT    | AHDINet | Ours        |
> |:---:|:-----------------:|:-----:|:------:|:----:|:-------:|:----:|:----:|
> | 0.00  | 35.05  | 35.47    | 36.46   | 36.58  | 36.67  | 37.09    | **37.61**   |
> | 0.01  | 35.05  | 35.12    | 36.36   | 36.58  | 35.62   | 37.06    | **37.58**   |
> | 0.02  | 35.05  | **32.64** | 36.09   | 36.57  | **34.97** | 36.86    | **37.54**   |
> | 0.03  | 35.05  | -        | 35.80   | 36.55  | -       | 36.47    | **37.45**   |
> | 0.04  | 35.05  | -        | 35.52   | 36.53  | -       | 35.97    | **37.31**   |
> | 0.05  | 35.05  | -        | 35.24   | 36.48  | -       | 35.45    | **37.12**   |
> | 0.06  | 35.05  | -        | **34.98** | 36.41  | -       | **34.96** | **36.85**   |
> | 0.07  | 35.05  | -        | -       | 36.30  | -       | -        | **36.52**   |
> | 0.08  | 35.05  | -        | -       | 36.14  | -       | -        | **36.31**   |
> | 0.09  | 35.05  | -        | -       | 35.91  | -       | -        | **36.13**   |
> | 0.10  | 35.05  | -        | -       | 35.62  | -       | -        | **35.70**   |
> | 0.11  | 35.05  | -        | -       | 35.28  | -       | -        | **35.27**   |
> | 0.12  | *35.05*| -        | -       | **34.95** | -     | -        | **34.84**   |
> ----------------------------------------
>
> ---
> Across under-reporting, temporal jitter, and noise, several baselines remain stable under mild perturbations, but most suffer rapid degradation as the perturbation increases. **RED maintains stable and superior performance across the entire perturbation range**, consistently outperforming all baselines and remaining above the image-only baseline even under heavy degradations.

---

> ### Author Response · Authors · 2025-11-24
>
> # W3/Q4 — Theoretical Details Behind the Robustness of RPS and MRM
>
> Thank you for raising this insightful question regarding the theoretical foundations of how RPS and MRM contribute to robustness. While the underlying details of RPS is presented across multiple sections (Introduction, Sections 3.1–3.3), their distributed placement may obscure the overall theoretical flow. Besides, the reason of three switching strategy is supplemented. Below, we reorganize the key mechanisms into a clearer explanation.
>
> ## (1) RPS: Robustness Through Threshold-Space Perturbation
>
> As derived in Section 3.1, event generation follows the contrast-triggering mechanism:
>
> $$
> e = \mathbb{I}(|\Delta L| > \theta),
> $$
>
> where $\Delta L$ is the log-intensity change and $\theta$ is the contrast threshold. Different thresholds induce different triggering distributions $p(e \mid \theta)$, affecting event density, temporal patterns, and noise levels. Thus, a real DVS effectively samples the same underlying motion field under a *family* of thresholds.
>
> RPS explicitly simulates this mechanism during training by perturbing $\theta$:
>
> $$
> \theta \sim \mathcal{T}, \qquad \tilde{e} \sim p(e \mid \theta),
> $$
>
> where $\mathcal{T}$ represents realistic threshold variations. The feature extractor $F$ is therefore optimized to remain stable across this threshold space:
>
> $$
> \min_F \ \mathbb{E}_{\theta \sim \mathcal{T}} \big|F(e(\theta)) - F^*\big|.
> $$
>
> This matches the intuition in our paper: **RPS prevents the model from over-specializing to a single threshold and instead promotes smooth, threshold-invariant feature responses**, effectively acting as a *data-space smoothing* operation that enhances robustness to under-reporting and noise.
>
> --------------------------------------------------------
>
> ## (2) MRM: Robustness Through Structured Representation Decoupling
>
> As shown in Section 3.2, let the intermediate features from the image and event branches be
> $F_{ei}^I,\, F_{ei}^E \in \mathbb{R}^{B \times N \times H \times W}$.
>
> The modality-specific encoders in MRM operate directly on these intermediate features:
> $$ F_{sem}^I = E_{img}(F_{ei}^I), $$
> $$ F_{mot}^E(\theta) = E_{evt}(F_{ei}^E(\theta)). $$
>
> The cross-modal component (Section 3.2 in the paper) produces
> $$ F_{cross}^C(\theta) = G_{cross}\big(F_{sem}^I,\; F_{mot}^E(\theta)\big). $$
>
> The final representation used for deblurring is the feature-space decomposition
> $$ F_{tot}(\theta) = F_{sem}^I \oplus F_{mot}^E(\theta) \oplus F_{cross}^C(\theta). $$
>
> Here, $\oplus$ denotes the decomposition of the feature space after modality-specific encoding and cross-modal reasoning.
>
> ---
>
> Consider two thresholds $\theta_1$ and $\theta_2$.
>
> Since the image branch does not depend on the DVS threshold,
> $$ F_{sem}^I(\theta_1) = F_{sem}^I(\theta_2). $$
>
> For the event-motion features, the dependence on $\theta$ comes from the event tensor:
> $$ F_{mot}^E(\theta_i) = E_{evt}\big(F_{ei}^E(\theta_i)\big), \quad i \in \{1,2\}. $$
>
> If $E_{evt}$ is $L_{evt}$-Lipschitz, then
> $$
> \big\| F_{mot}^E(\theta_1) - F_{mot}^E(\theta_2) \big\|
> \le
> L_{evt} \, \big\| F_{ei}^E(\theta_1) - F_{ei}^E(\theta_2) \big\|.
> $$
>
> For the cross-modal term,
> $$
> F_{cross}^C(\theta_i) = G_{cross}\big(F_{sem}^I\; F_{mot}^E(\theta_i)\big).
> $$
>
> If $G_{cross}$ is $L_{cross}$-Lipschitz in the motion feature, then
> $$
> \big\| F_{cross}^C(\theta_1) - F_{cross}^C(\theta_2) \big\|
> \le
> L_{cross} \, \big\| F_{mot}^E(\theta_1) - F_{mot}^E(\theta_2) \big\|.
> $$
>
> Combining the two inequalities gives
> $$
> \big\| F_{tot}(\theta_1) - F_{tot}(\theta_2) \big\|
> \le
> \big\| F_{mot}^E(\theta_1) - F_{mot}^E(\theta_2) \big\|
> +
> \big\| F_{cross}^C(\theta_1) - F_{cross}^C(\theta_2) \big\|,
> $$
> while the semantic feature $F_{sem}^I$ remains unchanged.
>
> This shows that threshold-induced variations affect only the motion-related feature $F_{mot}^E$ and its induced change in the cross-modal feature $F_{cross}^C$, while the semantic backbone $F_{sem}^I$ is invariant. Combined with RPS, which exposes the model to different $\theta$ during training, this leads to a representation robust to event under-reporting.
>
>
> ## (3) Combined Mechanism: Complementary Robustness in Input and Representation Spaces
>
> RPS and MRM provide robustness at two complementary levels:
>
> - **RPS broadens the range of event distributions encountered during training**, improving input-level tolerance to threshold-induced degradation.
> - **MRM constrains how threshold-induced noise propagates through the representation**, ensuring that unstable event cues cannot corrupt semantic features.
>
> Together, these mechanisms yield a representation in which **event-derived motion cues enhance deblurring while semantic stability is preserved**, enabling RED to remain robust under real-world threshold variations and degraded event streams.

---

> ### Author Response · Authors · 2025-11-24
>
> #  Q5 — Combining RPS with noise-injection or adversarial training
>
> Thank you for the constructive suggestion. We conducted additional experiments to investigate whether combining RPS with noise-injection training or adversarial training could provide further robustness improvements. Before presenting the results, we note that GoPro, HighREV, and REVD already contain natural degradations such as  motion blur. The results are summarized below.
>
> ---
>
> ## RPS with Noise-Injection Training
>
> From **Table NI-UR**, the performance of “Ours + Noise” remains almost identical to the original model under all under-reporting ratios. From this we can infer that：
> 1) Noise-injection does not interfere with the stability of RPS under under-reporting, confirming that RPS already captures the dominant degradation characteristics.
> 2) Under noise degradation, Ours + Noise yields slight improvements, suggesting that noise-injection can better handle purely noisy inputs without affecting RPS.
> ---
>
> **Table NI-UR: Performance of RPS with Noise-Injection under Under-Reporting**
>
> | **UR** | **Ours** | **Ours + Noise** |
> |:------:|:--------:|:----------------:|
> | 0    | **37.63 / 0.9802** | 37.59 / 0.9801 |
> | 0.05 | **37.55 / 0.9800** | 37.51 / 0.9799 |
> | 0.10 | **37.45 / 0.9797** | 37.42 / 0.9797 |
> | 0.15 | **37.34 / 0.9794** | 37.32 / 0.9794 |
>
> ---
>
> **Table NI-Noise: Performance of RPS with Noise-Injection under Noise Degradation**
>
> | **Noise Ratio** | **Ours** | **Ours + Noise** |
> |:---------------:|:--------:|:----------------:|
> | 0    | **37.63 / 0.9802** | 37.59 / 0.9801 |
> | 0.01 | **37.58 / 0.9799** | 37.5245 / 0.9799 |
> | 0.02 | 37.54 / 0.9796 | **37.5229 / 0.9799** |
> | 0.03 | 37.45 / 0.9791 | **37.5202 / 0.9798** |
> | 0.04 | 37.31 / 0.9783 | **37.5164 / 0.9798** |
> | 0.05 | 37.12 / 0.9771 | **37.5115 / 0.9798**|
>
> ---
>
> ## RPS with Adversarial Training(AT)
> In contrast, combining RPS with adversarial training consistently reduces performance across all conditions (Tables AT-UR and AT-Noise). We can infer that:
> 1) Ours + AT shows clear drops at every under-reporting and noise level.
> 2) This suggests that adversarial training biases the model toward texture generation rather than stable reconstruction, which negatively impacts event-guided deblurring.
> Adversarial training does not complement RPS and instead reduces base reconstruction quality, making it unsuitable for robust event-based deblurring.
>
> ---
> **Table AT-UR: Performance of RPS with Adversarial Training under Under-Reporting**
>
> | **UR** | **Ours** | **Ours + AT** |
> |:------:|:--------:|:-------------:|
> | 0    | **37.63 / 0.9802** | 37.06 / 0.9760 |
> | 0.05 | **37.55 / 0.9800** | 36.97 / 0.9760 |
> | 0.10 | **37.45 / 0.9797** | 36.88 / 0.9757 |
> | 0.15 | **37.34 / 0.9794** | 36.77 / 0.9753 |
>
> ---
>
> **Table AT-Noise: Performance under Noise Degradation**
>
> | **Noise Ratio** | **Ours** | **Ours + AT** |
> |:---------------:|:--------:|:-------------:|
> | 0    | **37.63 / 0.9802** | 37.06 / 0.9760 |
> | 0.01 | **37.58 / 0.9799** | 36.89 / 0.9752 |
> | 0.02 | **37.54 / 0.9796** | 36.86 / 0.9748 |
> | 0.03 | **37.45 / 0.9791** | 36.79 / 0.9742 |
> | 0.04 | **37.31 / 0.9783** | 36.68 / 0.9729 |
> | 0.05 | **37.12 / 0.9771** | 36.47 / 0.9708 |

---

### Official Review · Reviewer_Gsta · 2025-11-01

**Soundness:** 3
**Presentation:** 2
**Contribution:** 3
**Rating:** 4
**Confidence:** 4

**Summary:**

The paper deals with the issue of under-reporting of events for event-based motion deblurring task, and proposes a network named RED. The paper introduces Robustness-Oriented Perturbation Strategy (RPS) to enhance the robustness and adaptability of RED to real-world conditions. A Modality-specific Representation Mechanism(MRM) is designed to explicitly model semantic understanding, motion priors, and cross-modality correlations from blurry images and events.  Two interactive modules MSEM/ESEM are presented to enhance motion-sensitive areas in blurry images and inject semantic context into under-reporting event representations.

**Strengths:**

1. The topic of handling under-reporting events in various DVS thresholds is an interesting topic for event-based deblurring. Bad event streams may indeed influence the fusion of two modalities.
2. It seems that the method can generalize better than others under high under-reporting ratio.
3. The figures and tables in the paper are well-organized.

**Weaknesses:**

1. The structure of the manuscripts is not good. Section 3.3 is too short to introduce the proposed modules. It appears that these two modules may have been combined in a way that gives the impression of being designed primarily to fulfill the workload requirement, rather than for a clear technical motivation.
2. The novelty of the MRM, as well as MSEM and ESEM, is not satisfying. The attention operations in the MRM are common modules in the field of the Transformer. Besides, there is no need for the full names of these modules to be so complicated.
3. The results in Figure 6 are not satisfying. The RED brings no visual improvement compared to other methods on this figure.
4. In real scenarios, are there situations when the under-reporting issue such high as the paper's experiments conducted?

**Questions:**

As listed in the "Weaknesses".

---

> ### Author Response · Authors · 2025-11-24
>
> # W1. Presentation of Section 3.3
> ----------
>
> ### **(1) Clarification on Section Length and Content Placement**
>
> We thank the reviewer for the insightful comments. The conciseness of Section 3.3 in the initial submission was mainly due to the 9-page limit, under which we prioritized presenting extensive robustness analyses and comparisons under diverse degradations. As indicated in the original supplementary material (Sec. A.3, L764–L799), the complete architectural details, structural diagrams, and implementation specifications of MSEM and ESEM were already provided there. **In the revised 10-page version, we expand Section 3.3 to include additional module descriptions(Page 6), information-flow explanations(Figure 4), and the corresponding visual evidence(Figure 3)**.
>
> ----------
>
> ### **(2)Technical Motivation Behind MSEM and ESEM**
>
> Regarding the concern that the two modules may appear combinational rather than technically motivated, we would like to clarify the core rationale behind their design. Both modules directly arise from two fundamental discrepancies we observed in the RPS formulation and in real event distributions. Under high contrast thresholds, event streams retain motion cues yet suffer from severe semantic under-reporting, leaving only sparse edge-like responses. Conversely, the image modality preserves rich semantics but loses reliable motion structure when heavily blurred. Direct fusion of uncorrected features from the two modalities therefore amplifies their distributional mismatch and leads to representational degradation. Since MRM has already established a semantic–motion decoupling, what is required at this stage is not further mixing, but a targeted, directional compensation that preserves the disentangled structure.
>
> ----------
>
> ### **(3)Purpose and Placement of Directional Compensation**
>
> MSEM transfers event-derived high-frequency motion cues to the image branch, enabling its features to recover motion structures suppressed by blur. ESEM injects high-level semantic priors from the image branch into the sparse event branch, reinforcing structural completeness when severe under-reporting occurs. The intention of both modules is not to recombine modalities, but to perform controlled, single-direction cross-modal correction within their respective semantic/motion subspaces. This is why both modules are placed early in the feature extraction stage, without altering the semantic–motion division of labor established by MRM in later layers.
>
> ----------
>
> ### **(4)Empirical Evidence Supporting the Module Behavior**
>
> To validate this behavior, we visualize the activation maps of both branches before and after MSEM/ESEM in **Figure 3 and Figure 11**, including: (1) image-encoder features before and after MSEM, (2) image-branch decoder features, (3) event-encoder features before and after ESEM, and (4) event-branch decoder features. The visualizations reveal the following patterns:
> - **Before MSEM**, image activations exhibit broad semantic layouts, while event activations focus on high-motion regions.
> - **After MSEM**, the image branch preserves its semantic structure, but exhibits enhanced responses exactly at event-indicated motion areas.
> - **After ESEM**, the event branch receives semantic reinforcement from the image branch, producing motion representations with more complete contours and improved structural continuity.
> - **In the decoder**, the image modality consistently governs global structure and coherence, while the event modality strengthens local detail recovery and edge transitions.
>
> These observations demonstrate that the primary functional separation between semantic and motion cues remains intact throughout the pipeline, and that MSEM/ESEM operate as intended: **providing directed cross-modal compensation without collapsing the disentangled representation space**.

---

> ### Author Response · Authors · 2025-11-24
>
> # W2. Naming of MRM, MSEM, and ESEM
>
> Thank you for the comment. We would like to clarify that the novelty of MRM, MSEM, and ESEM does not lie in proposing new attention operators, but in redesigning the cross-modal representation pipeline specifically around the characteristics of event under-reporting. The contribution is structural and problem-driven rather than operator-driven.
>
> First, while attention is indeed a common component in Transformer architectures, the motivation of MRM is fundamentally different from standard cross-modal attention. Existing event-guided deblurring methods typically follow a “joint encoding + direct fusion” paradigm, implicitly assuming that blurry image features and event features lie in compatible distributions. Our analysis shows that this assumption breaks when under-reporting occurs: event features suffer from structural semantic loss, and naive fusion leads to semantic–motion contamination and representational collapse. MRM therefore introduces a new representational decomposition from **semantic**, **motion**, and **cross-modal** subspaces, which explicitly prevents such contamination by enforcing functional separation prior to interaction. This structured decomposition has not appeared in prior event-based deblurring work.
>
> To further support this point, we include an ablation study where each component of MRM is selectively replaced with a modality-agnostic self-attention module while keeping feature dimensionality unchanged (following **Yuan et al. 2021 [1]**). The results are displayed here for clarity:
>
> ----------
>
> **Table: Ablation study on the semantic, motion, and cross-modal components of MRM**
>
> | **Semantic** | **Motion** | **Cross** | **PSNR (↑)** | **SSIM (↑)** |
> |--------------|------------|------------|---------------|---------------|
> | ✗ | ✗ | ✗ | 25.77 | 0.864 |
> | ✗ | ✓ | ✓ | 34.99 | 0.964 |
> | ✓ | ✗ | ✓ | 35.04 | 0.966 |
> | ✓ | ✓ | ✗ | 37.14 | 0.978 |
> | ✓ | ✓ | ✓ | **37.63** | **0.980** |
>
> This ablation demonstrates three key observations:
> (1) Replacing all specialized components with generic self-attention results in an 11.86 dB PSNR drop, indicating that processing events and images indiscriminately severely harms both feature extraction and feature interaction.
> (2) Replacing only the image-branch or event-branch attention causes 2.64 dB and 2.59 dB drops respectively, while replacing the cross-modal attention still yields a noticeable 0.49 dB degradation.
> (3) Taken together, these results confirm that MRM is essential for capturing distinct semantic content and motion-sensitive regions, and for enabling complementary cross-modal fusion that is robust to under-reporting.
>
> Second, the novelty of MSEM and ESEM lies in their **directional and asymmetric compensation functions**, rather than architectural complexity. Their names intentionally highlight this functional asymmetry: MSEM propagates event-derived motion cues to the image branch to restore motion suppressed by blur, while ESEM injects image-derived semantic priors into sparse event features to recover structural completeness under severe under-reporting. These two compensation paths address opposite failure modes and are therefore not redundant. Detailed explanations of MSEM and ESEM are provided in the following response.
>
> [1] Li Yuan, Yunpeng Chen, Tao Wang, Weihao Yu, Yujun Shi, Zi-Hang Jiang, Francis EH Tay, Jiashi Feng, and Shuicheng Yan. Tokens-to-token vit: Training vision transformers from scratch on imagenet. In Proceedings of the IEEE/CVF international conference on computer vision, pp.558–567, 2021.

---

> ### Author Response · Authors · 2025-11-24
>
> # W3. Visualization Comparison in Figure 6
>
> Thank you for the comment. We would like to clarify that Figure 6 corresponds to challenging real-world scenes, where severe event degradations jointly limit the performance of all event-guided deblurring methods. In such scenarios, several existing approaches even produce results worse than the blurry input. Within this context, we highlight three observations showing that RED still exhibits clear advantages in Figure 6 (Figure 7 in the revised manuscript).
>
> First, RED provides visible improvements in structural integrity and texture quality. As noted by the reviewer in the Figure 6, alternative methods suffer from oversmoothing, structural tearing, or noise-induced artifacts in regions such as tree trunks, grass, and road boundaries. RED, in contrast, reconstructs more continuous contours, clearer grass textures, and cleaner edge transitions.
>
> Second, the quantitative results consistently support that RED is not “providing no improvement.” On the same REVD setting, Table 2(b) shows that RED outperforms all competitors in both PSNR and SSIM, and crucially never falls below the blurry input.
>
> Third, additional real-world examples in Appedix (**Fig.17, Fig.18 in the revised manuscript**) further demonstrate RED’s advantages under complex event degradations. In HighREV, multiple methods fail to exceed the blurry input or produce misaligned edges and distorted textures; in REVD, some methods show over-sharpening artifacts or strong hallucinations. RED consistently yields sharper boundaries, more coherent textures in hair and clothing, and fewer artifacts across diverse scenes, indicating stronger robustness to real sensor degradation.
>
> To make these observations more accessible, we have added detailed figure captions for all visual comparisons in the revised manuscript (**Figures 7, 8, 17, and 18**), clarifying the structural and textural advantages of RED in challenging real-world scenarios.
>
> ---------------------------
>
> # W4 Under-reporting in Real Scenes
>
> Thank you for the question. We clarify that the levels of under-reporting modeled in our experiments are not artificially extreme. Instead, they reflect inherent and widely documented limitations of current event cameras. We provide explanation from sensing-theory, quantitative statistics on real event streams, and visual evidence.
>
> ### (1) Under-reporting inevitably occurs in many real-world conditions.
> As discussed in Section 3.1, event generation depends on whether the log-intensity change exceeds the contrast threshold. In practice, brightness changes in low-light environments, low-texture areas, and strong backlight often fail to cross this threshold, resulting in missing events. These sensing limitations and the resulting under-reporting behavior are well documented in the survey by **Gallego et al. 2020 [1]**.
>
> ### (2) Approximate quantitative statistics from real REVD data.
> Since ground-truth event density is unavailable, we estimate event coverage by computing how often high-gradient image regions (where events might occur) exhibit zero or only one event. Image gradients serve as a strong proxy for brightness changes; missing events in such regions provide a reliable indication of under-reporting.
>
> **Table: Ratio of severely under-reported high-gradient patches across REVD sequences**
>
> | **Scene** | **Samples** | **Ratio (grad threshold)** |
> |-----------|-------------|----------------------------|
> | 1 | 266 | 0.198870 |
> | 2 | 239 | 0.099899 |
> | 3 | 261 | 0.154541 |
> | 4 | 256 | 0.083659 |
> | 5 | 254 | 0.050801 |
> | 6 | 283 | 0.045483 |
>
> From these measurements, we observe that **all six REVD sequences exhibit noticeable under-reporting**, with ratios ranging from **0.04 to 0.20** in high-gradient regions. These values closely match the degradation levels modeled in our experiments, confirming that such under-reporting is not an artificial setting but a property of real sensor behavior, especially in dynamic scenes.
>
> ### (3) Real-world visual comparisons.
> Furthermore, the experimental results in **Table 3** and the visual comparisons in **Figures 7, 8, 17, and 18** demonstrate that event degradation in real conditions is significant enough to cause existing methods to fail, sometimes producing outputs worse than the blurry inputs. These visualizations corroborate that real-world deblurring is indeed challenged by severe under-reporting, matching our problem formulation.
>
> In summary, under-reporting is an inherent and unavoidable characteristic of  event sensors. Its severity in real scenes is well aligned with the conditions modeled in our experiments, making RED’s design not only necessary but practically motivated. We have added a brief discussion in the revised related work section and included our REVD statistics in the supplementary material.
>
> [1] Guillermo Gallego, et al. Event-based vision: A survey. IEEE transactions on pattern analysis and machine intelligence, 44(1):154–180, 2020.

---

### Official Review · Reviewer_3XiY · 2025-11-02

**Soundness:** 3
**Presentation:** 3
**Contribution:** 3
**Rating:** 4
**Confidence:** 3

**Summary:**

This paper addresses the unstable deblurring performance caused by the "noise-under-reporting" trade-off of event cameras, proposing a Robust Event-guided Deblurring (RED) network. The core innovations include: designing a Robustness-Oriented Perturbation Strategy (RPS) to simulate event under-reporting patterns under different DVS thresholds, enhancing the model's adaptability to real-world scenarios; proposing a Modality-specific Representation Mechanism (MRM) to disentangle image semantic and event motion features, avoiding mixed representations; introducing two bidirectional interaction modules (MSEM and ESEM) to achieve complementary fusion of motion priors and semantic information. Experiments show that RED achieves state-of-the-art performance on both synthetic and real-world datasets, and RPS can be used as a plug-in to improve the robustness of other models. This method effectively solves the core pain point of event-guided motion deblurring, providing a robust and practical solution for deblurring in high-dynamic scenarios.

**Strengths:**

1.This paper systematically addresses the "noise-under-reporting" trade-off of event cameras. The RPS strategy innovatively simulates the physical mechanism of real event acquisition, the MRM realizes the disentanglement of modal features, breaking through the limitation of mixed features in existing methods, and the design of bidirectional interaction modules is highly targeted.
2. The experimental design is comprehensive, covering multiple synthetic and real-world datasets, verifying the method's robustness under different under-reporting ratios. Ablation experiments detailedly validate the necessity of each component including RPS, MRM, and MSEM/ESEM. The plug-and-play generality of RPS is verified by integrating it into other models, with highly credible conclusions.
3. The paper has a coherent structure, progressing layer by layer from problem analysis, method design to experimental verification. Framework diagrams and performance curves intuitively show the method flow and advantages. Technical details (such as the probabilistic modeling of RPS and the attention mechanism of MRM) are elaborated in detail, and the literature review is comprehensive, facilitating the understanding of the research background.

**Weaknesses:**

1. Suboptimal computational efficiency: The parameter count (19.2M) and computational complexity (637.45G FLOPs) of RED are relatively high. Compared with lightweight models (e.g., EFNet: 7.73M parameters, 379.43G FLOPs), there is a lack of efficiency-performance trade-off analysis.
Insufficient depth of modal interaction mechanism: The feature fusion methods of MSEM and ESEM are relatively simple (such as element-wise multiplication and concatenation), and do not consider the dynamic adaptation between the degree of event under-reporting and image blur intensity. The cross-modal attention of MRM does not distinguish the differential adaptation of different motion types (e.g., rigid body motion, non-rigid body motion).
2.Incomplete coverage of extreme scenarios: Experiments do not involve extreme scenarios such as complete event loss, coexistence of strong noise and high under-reporting, and ultra-high-definition images (e.g., 4K). The adaptability of the method to different event camera models (different DVS threshold characteristics) is not evaluated.
3. Limited depth of core innovation: "Perturbation training to improve robustness" and "modality-specific feature extraction" are already mature ideas in the fields of image restoration and cross-modal fusion. The innovation of RED is more about the adaptive combination of existing ideas in event-guided deblurring tasks, lacking breakthrough paradigm innovation.

**Questions:**

1. What is the performance of RED in high-noise scenarios with low DVS thresholds? Can supplementary comparative experiments under different noise intensities be provided to illustrate its advantages in noise robustness compared to methods such as EFNet and AHDINet?

2. Can the feature fusion weights of MSEM and ESEM be dynamically adjusted according to the degree of event under-reporting and image blur intensity? How adaptable is MRM to different motion types (rigid/non-rigid), and is targeted optimization required?

---

> ### Author Response · Authors · 2025-11-24
>
> # Q1/W1.4(Part 1)  Performance under extreme scenarios
>
> Thank you for raising this question. We first clarify our definition of *extreme scenarios*. In event-based deblurring, a method is considered to **collapse** once its performance falls below the image-only baseline, as the event stream no longer provides meaningful motion cues. Therefore, our objective is not to construct physically unrealistic corner cases, but to evaluate robustness across the full degradation spectrum where event signals remain informative.
>
> Following this principle, we conduct a comprehensive robustness evaluation along three degradation dimensions, including under-reporting, temporal jitter, and background noise, each progressively extended toward the boundary at which most existing methods collapse. The detailed results are shown in Tables 8, 9, and 10, where the image-only performance serves as a reference baseline. Below are comparisons of PSNR.
>
> ---
>
> ## **(a) Under-Reporting**
>
> From Table 1, baselines degrade quickly as the ratio increases, some reaching the image-only baseline around UR ≈ 0.1–0.15. RED maintains the most stable behavior up to UR = 0.6.
>
> **Table 1. Performance under Under-Reporting (PSNR)**
>
> ----------------------------------------
> | UR    | DSTN(no event)   | EFNet  | STCNet | TRMD  | AHDINet | MAT    | Ours        |
> |-------|---------|---------|---------|---------|----------|---------|-------------|
> | 0.00  | 35.05   | 35.47   | 36.46   | 36.58   | 37.09    | 36.67   | **37.63**   |
> | 0.05  | 35.05   | 35.25   | 35.93   | 35.77   | 35.78    | 35.77   | **37.55**   |
> | 0.10  | 35.05   | **34.92** | 35.39   | 35.05   | **34.77** | 35.53   | **37.45**   |
> | 0.15  | 35.05   | -       | **34.84** | **34.42** | -        | 35.20   | **37.34**   |
> | 0.20  | 35.05   | -       | -       | -       | -        | **34.81** | **37.21**   |
> | 0.25  | 35.05   | -       | -       | -       | -        | -       | **37.06**   |
> | 0.30  | 35.05   | -       | -       | -       | -        | -       | **36.89**   |
> | 0.35  | 35.05   | -       | -       | -       | -        | -       | **36.67**   |
> | 0.40  | 35.05   | -       | -       | -       | -        | -       | **36.40**   |
> | 0.45  | 35.05   | -       | -       | -       | -        | -       | **36.07**   |
> | 0.50  | 35.05   | -       | -       | -       | -        | -       | **35.63**   |
> | 0.55  | 35.05   | -       | -       | -       | -        | -       | **35.06**   |
> | 0.60  | *35.05* | -       | -       | -       | -        | -       | **34.34**   |
>
> ---
> ## **(b) Temporal Jitter**
>
> From Table 2, mild jitter (0.1–0.2) affects only a few methods, while stronger jitter leads to rapid performance drop for most baselines. RED remains consistently superior across all jitter levels.
>
> **Table 2. Performance under Temporal Jitter (PSNR)**
>
> ----------------------------------------
> | Jitter | DSTN(no event)  | EFNet   | STCNet | TRMD  | MAT    | AHDINet | Ours        |
> |--------|--------|----------|---------|--------|---------|----------|-------------|
> | 0.1    | 35.05  | 35.44    | 35.71   | 36.46  | 35.95   | 36.54    | **37.50**   |
> | 0.2    | 35.05  | 35.38    | **34.43** | 36.17  | 35.91   | 36.23    | **37.18**   |
> | 0.3    | 35.05  | 35.29    | -       | 35.80  | 35.85   | 35.95    | **36.84**   |
> | 0.4    | 35.05  | 35.15    | -       | 35.39  | 35.76   | 35.56    | **36.39**   |
> | 0.5    | 35.05  | **34.95** | -       | 34.82  | 35.62   | 35.16    | **35.92**   |
> | 0.6    | 35.05  | -        | -       | -      | 35.42   | **34.81** | **35.51**   |
> | 0.7    | *35.05*| -        | -       | -      | **35.03** | -        | **35.01**   |
> ----------------------------------------

---

> ### Author Response · Authors · 2025-11-24
>
> # W2/Q2(Part 2) — Performance under extreme scenarios
>
> ## **(c) Background Noise**
>
> From Table 3, most baselines collapse as noise approaches 0.1, while RED maintains the smoothest and highest overall performance.
>
> **Table 3. Performance under Background Noise (PSNR)**
>
> ----------------------------------------
> | Noise | DSTN(no event)  | EFNet   | STCNet | TRMD  | MAT    | AHDINet | Ours        |
> |-------|--------|----------|---------|--------|---------|----------|-------------|
> | 0.00  | 35.05  | 35.47    | 36.46   | 36.58  | 36.67   | 37.09    | **37.61**   |
> | 0.01  | 35.05  | 35.12    | 36.36   | 36.58  | 35.62   | 37.06    | **37.58**   |
> | 0.02  | 35.05  | **32.64** | 36.09   | 36.57  | **34.97** | 36.86    | **37.54**   |
> | 0.03  | 35.05  | -        | 35.80   | 36.55  | -       | 36.47    | **37.45**   |
> | 0.04  | 35.05  | -        | 35.52   | 36.53  | -       | 35.97    | **37.31**   |
> | 0.05  | 35.05  | -        | 35.24   | 36.48  | -       | 35.45    | **37.12**   |
> | 0.06  | 35.05  | -        | **34.98** | 36.41  | -       | **34.96** | **36.85**   |
> | 0.07  | 35.05  | -        | -       | 36.30  | -       | -        | **36.52**   |
> | 0.08  | 35.05  | -        | -       | 36.14  | -       | -        | **36.31**   |
> | 0.09  | 35.05  | -        | -       | 35.91  | -       | -        | **36.13**   |
> | 0.10  | 35.05  | -        | -       | 35.62  | -       | -        | **35.70**   |
> | 0.11  | 35.05  | -        | -       | 35.28  | -       | -        | **35.27**   |
> | 0.12  | *35.05*| -        | -       | **34.95** | -     | -        | **34.84**   |
> ----------------------------------------
>
> ---
> Across under-reporting, temporal jitter, and noise, several baselines remain stable under mild perturbations, but most suffer rapid degradation as the perturbation increases. **RED maintains stable and superior performance across the entire perturbation range**, consistently outperforming all baselines and remaining above the image-only baseline even under heavy degradations.

---

> ### Author Response · Authors · 2025-11-24
>
> # W1.1 Computational Efficiency
>
> Thank you for your consideration of efficiency. We would like to emphasize that RED is designed primarily to address *robustness under severe event degradation*, rather than to pursue lightweight architectures. The core challenge in event-based deblurring is not FLOPs alone, but the ability to maintain stable reconstruction when events are heavily under-reported or corrupted by noise. As shown in Table 4/5 of our manuscript, lightweight CNN-based approaches such as **EFNet** (CVPR 2024) and **STCNet** (ECCV 2022) exhibit clear performance collapse as UR increases or noise intensifies. In contrast, RED maintains **35.63 dB** at UR = 0.5, while several lightweight baselines fall below 30 dB or even deteriorate to worse-than-blurry outputs. These results directly demonstrate that efficiency cannot replace robustness for this task.
>
> The trade-off between computational cost and robustness is also reflected when comparing Transformer-based and CNN-based designs. CNN-based models (EFNet, STCNet, **AHDINet** ICCV 2023) are relatively computationally efficient, but their temporal generalization is limited; their feature extractors rely heavily on dense, clean events, and degrade sharply once event sparsity increases. Transformer-based backbones (such as **TRMD** CVPR 2024, **MAT** ICCV 2023, and RED) incur higher FLOPs, but offer substantially stronger resilience to event disturbances due to their global modeling capability. Importantly, RED achieves the *best overall performance across all UR conditions* while maintaining **lower parameter count than MAT** and comparable complexity to TRMD.
>
> **Table. Comparison of backbone type, parameter count, and PSNR under increasing under-reporting**
>
> | **Method**     | **Backbone**                   | **Params (M)** | **FLOPs (G)** | **UR = 0 PSNR** | **UR = 0.5 PSNR** |
> |----------------|--------------------------------|----------------|---------------|------------------|---------------------|
> | DSTN           | CNN (image-only)               | 7.45           | 168.29        | 35.05            | 35.05              |
> | EFNet          | CNN                             | 7.73           | 379.43        | 35.47            | 31.03              |
> | STCNet         | CNN                             | 8.54           | 669.65        | 36.46            | 30.96              |
> | TRMD           | Transformer                     | 19.26          | 113.02        | 36.58            | 31.41              |
> | AHDINet        | CNN                 | 10.60          | 288.04        | 37.09            | 29.64              |
> | MAT            | Transformer                     | 20.73          | 749.91        | 36.67            | 31.24              |
> | **RED (Ours)** | Transformer (modality-aware)    | **19.20**      | **637.45**    | **37.63**        | **35.63**          |
>
>
> These results highlight a key point: **RED achieves the best performance–robustness balance among Transformer-based methods, while remaining significantly more robust than lightweight CNN baselines under degraded events**. The additional computational cost is therefore not arbitrary, but a necessary design choice supported by empirical evidence across all UR settings.
>
> --------------------------------------------------------------------
> # W1.3 — Adaptability of MRM to Rigid vs. Non-Rigid Motion
>
> Thank you for raising this point. We clarify that distinguishing between rigid and non-rigid motion is **not necessary** for event-based restoration, because in the event sensing model the two motion types are **not separable at the signal level**.
>
> In the standard event camera model, an event is triggered when the log-intensity change at a pixel exceeds a contrast threshold: $ E(x,y,t) = \text{sign}(\Delta L(x,y,t)). $
>
> This formulation shows that event signals depend solely on the **local temporal gradient of log-intensity**, not on the physical nature of the motion (rigid vs. non-rigid). Both rigid-body motion and non-rigid deformation produce events through the *same* mechanism, as pixel-wise brightness change, differing only in spatial distribution or magnitude, not in mathematical form. Consequently, existing event-based deblurring methods (EFNet, AHDINet, MAT, TRMD, etc.) do not explicitly model motion types, because event data fundamentally does not provide such separability.
>
> MRM follows this established physical principle: its motion representation is derived from the same pixel-level temporal contrast changes. Therefore, MRM naturally accommodates both rigid and non-rigid motion without requiring separate optimization or motion-type modules. Besides, we demonstrates feature maps in four scenes in Figure 3 and 11, which demonstrates our MRM consistently highlights motion-sensitive regions according to the events.

---

> ### Author Response · Authors · 2025-11-24
>
> # W1.2/Q2.1 — MSEM and ESEM
>
> Thank you for the detailed comments for the interaction modules. We clarify the functional roles of the multiplicative weights in MSEM/ESEM and how they relate to event degradation.
>
> In both modules, the multiplicative weights in Figure 4 are not redundant operations but correspond to different levels of compensation.
> In **MSEM**, the first modulation
> $$
> \widehat{F}^{(1)}_E = F^{(i)}_E \odot S + F^{(i)}_E
> $$
> uses the event-driven saliency map **S** to highlight motion-dominant regions within the event stream. This is an *intra-event* enhancement stage that amplifies reliable motion cues and suppresses background noise.
>
> After concatenating with the image feature, we compute a second weight **γ**, which captures *cross-modal motion relevance* from the mixed representation. The refinement
> $$
> \widehat{F}^{(2)}_E = \gamma \odot \widehat{F}^{(1)}_E
> $$
> ensures that only the motion cues that remain meaningful after cross-modal inspection are propagated to the image branch. Therefore, **S determines where motion exists in the event domain, while γ determines which of these motion cues are beneficial for image refinement**. They serve complementary roles rather than repeating the same operation.
>
> In **ESEM**, the weight **β** plays the symmetric role from the image side. It encodes semantic relevance within the image feature and extracts stable semantic priors to compensate for structurally missing events. The fused representation is then processed by temporal attention (for event cues) and spatial attention (for semantic cues), producing the enriched event feature. Thus, β enhances the event branch precisely where semantic reinforcement is needed.
>
> An important point is that these modulations already exhibit **implicit, data-driven adaptation** to the severity of event under-reporting:
> - when events are sparse or unreliable, both S and γ naturally decrease in magnitude, reducing the influence of the event branch while β becomes more dominant;
> - when events are rich and reliable, S and γ naturally strengthen, and β becomes less influential.
> This built-in adaptivity arises directly from the feature distributions and avoids the instability associated with explicit dynamic gating, while maintaining the semantic–motion separation required by MRM.
>
> To validate the behavior of these modules, we include activation visualizations in **Figures 3 and 11** of the revised manuscript. They show that:
> (1) after MSEM, image features preserve semantic layouts while selectively strengthening responses in event-indicated motion areas;
> (2) after ESEM, the event branch receives semantic reinforcement, yielding more continuous structural contours;
> (3) throughout the pipeline, semantic and motion roles remain clearly separated.
> These observations confirm that MSEM and ESEM provide directional, interpretable compensation without collapsing the disentangled representation space.

---

> ### Author Response · Authors · 2025-11-24
>
> # W2 — Response to “Limited depth of core innovation”
>
> Thank you for the reviewer’s comments. We respectfully clarify that the main objective of RED is not to introduce a new attention operator, but to address a core and long-standing challenge in event-guided deblurring: achieving stable motion reconstruction under severe under-reporting and strong sensor noise. This problem has been consistently overlooked in prior works, and our contributions are driven by the physical behavior of event cameras rather than by architectural novelty for its own sake.
>
> ## 1. RPS is not generic perturbation training but a sensor-grounded formulation
> As derived in Section 3.1 of the paper, event triggering follows the contrast threshold model
> $\(|S + N_p + N_g| \ge \theta\)$.
> Increasing the threshold simultaneously suppresses false positives and discards weak motion signals, producing structured under-reporting that cannot be reproduced by common augmentations such as random masking or noise injection.
>
> RPS directly corresponds to this physical mechanism by modeling event survival probability and applying Bernoulli thinning. It is architecture-agnostic, parameter-free, and grounded in the sensor’s generative process. Therefore, its purpose and effect differ fundamentally from conventional augmentation strategies.
>
> ## 2. Modality-specific representation is essential under realistic under-reporting
> “Modality-specific feature extraction” in RED is not adopted as a generic design principle. It arises from a structural problem caused by under-reporting: event features lose semantic completeness in a non-random, signal-dependent manner. When naively fused with image features, this leads to semantic–motion contamination, which is the primary reason existing methods collapse under disrupted events.
>
> MRM, MSEM, and ESEM form a unified solution to this issue:
>
> - **MRM** explicitly decomposes semantic, motion, and cross-modal subspaces to prevent mutual interference;
> - **MSEM** strengthens motion-sensitive regions in the image pathway using the event’s high-frequency responses;
> - **ESEM** injects high-level semantic priors into sparse event maps to maintain coherent structure under severe under-reporting.
>
> These modules are not a combination of existing ideas but a cohesive system designed precisely to handle the degradation patterns dictated by the event-triggering mechanism. Ablation studies further show that each component is indispensable for maintaining robustness.
>
> ## 3. Summary
> RED’s contributions are not based on assembling known techniques. Both RPS and the modality-specific representation framework originate from the physical formulation of event degradation and address challenges that prior methods have not resolved. The system demonstrates consistent improvements across different degradation regimes, validating that the innovation lies in problem-driven, sensor-grounded design rather than architectural embellishment.

---

> ### Comment · Reviewer_3XiY · 2025-11-28
>
> Overall, the authors have effectively addressed my major concerns. The strengths of the paper, particularly in tackling the specific "noise-under-reporting" trade-off, are now much clearer. I am inclined to raise my score accordingly, and I support the acceptance of this paper.

---

> > ### Author Response · Authors · 2025-11-28
> >
> > Dear Reviewer  3XiY：
> >
> > Thank you for your comment and positive feedback.
> > We sincerely appreciate your valuable suggestions and are pleased to hear that our response has addressed your concerns. We are grateful for your willingness to raise the score and for your support of our work. We plan to open-source this pipeline as a contribution to the community.
> >
> > Sincerely,
> >
> > The Authors

---

### Author Response · Authors · 2025-11-30
**Rebuttal Progress and Key Clarifications**

**Dear Program Chairs, Senior Area Chairs, Area Chairs, and Reviewers,**

We sincerely thank all program chairs, senior area chairs, area chairs, and reviewers (**R1-3XiY, R2-Gsta, R3-gaMS, R4-bp3s**) for your time and dedication to ensuring a fair review process. We understand the significant additional workload caused by the recent OpenReview incident. To assist in your reassessment, we would like to provide a concise summary of our paper’s status and the consensus reached:

**1. Rebuttal Progress and Score Update**

We engaged in active discussions with all four reviewers and received constructive feedback.

- **Initial Scores:** [4, 4, 6, 6] (Mean: 5.0)
- **Post-Rebuttal Progress:**
  - **Reviewer 3XiY (Initial Score: 4)**, after our rebuttal, **explicitly raised their score and now supports acceptance**. This followed our new experiments resolving core concerns on the model's **robustness under multiple degradations** and the **"noise-under-reporting" trade-off**.
  - While other reviewers did not update their comments before Nov 27, the issues they raised were also comprehensively addressed in our responses and revised manuscript.

**2. Reviewer Consensus on Core Contributions**

There is a strong consensus among all reviewers (3XiY, Gsta, gaMS, bp3s) on the core contributions of our work:

- **Problem Significance:** Reviewers agree that the "noise-under-reporting" trade-off is a critical and long-standing problem in event-based deblurring.
- **Effective RPS Strategy:** Reviewers praise our RPS strategy as a practical, model-agnostic, and effective method for significantly improving model robustness.
- **Superior Performance:** The state-of-the-art performance of RED is acknowledged, especially its robustness under harsh conditions with high under-reporting rates.
- **Experimental Rigor:** All reviewers commend the comprehensive experimental design and rigorous ablation studies that validate the contribution of each component.

**3. Key Concerns Addressed During Rebuttal**

In response to reviewer concerns, we significantly enhanced the manuscript with substantial new work:

- **Robustness on noise and temporal jitter** To address the core concerns from **Reviewer 3XiY** and **Reviewer gaMS** about the model's robustness under multi-dimensional degradations, we added two sets of tests for **temporal jitter and background noise** beyond **high under-reporting**. The results (see Rebuttal comments) demonstrate that RED remains the most stable and top-performing method, even as other methods collapse due to poor event quality.
- **Clarified the Computational Overhead of RPS:** Responding to the specific query from **Reviewer gaMS**, we provided precise quantitative data. For an event tensor of size $(B,T,C,H,W)=(1,6,1,360,640)$, averaged over 100 runs, RPS introduces only **2.49M FLOPs** and roughly **0.71 ms** of runtime overhead at an under-reporting ratio of 0.2. We emphasize that RPS is a **parameter-free, plug-and-play, and model-agnostic** module.
- **Enhanced Design Explainability:** Following **Reviewer gaMS**'s suggestion, we added **visualizations (Figs. 3 & 11)** that visually confirm our "disentangle-then-fuse" design and the synergistic roles of our modules. This also allowed us to address **Reviewer Gsta**'s concerns by reorganizing content previously in the appendix, adding **detailed module descriptions on Page 6**, supplemented by a **detailed structural flowchart (Figure 4)** and **visual evidence (Figure 3)**.
- **Clarified RPS is More Than Augmentation:** For **Reviewer bp3s**, we clarified RPS is a physics-based simulation (in Lines 183-231), not random perturbation. Crucially, our experiments show that while RPS boosts other models (AHDINet, MAT), a significant performance gap to our full method remains. This gap proves that simply seeing degraded data is not enough. The reason is architectural: existing methods with symmetric backbones **lack the capability to effectively process corrupted events**. Our framework, however, offering a new perspective for this task: MRM **first disentangles** **and then fuses** modality-specific features, then MSEM/ESEM perform **targeted compensation**.

**4. Overall Summary**

Our work provides a robust solution to the critical "under-reporting" challenge in event cameras. Our proposed physics-based RPS strategy and "disentangle-then-fuse" framework are experimentally validated to effectively extract beneficial motion representations from corrupted events. The rebuttal process earned the explicit support of **Reviewer 3XiY (initial score: 4)**, has further strengthened our claims.

We hope this summary assists you in efficiently navigating our review history. **Thank you again for your hard work.**

Sincerely,

**Authors of Submission 6537**

---

### Meta-Review · Area_Chair_f1WD · 2026-01-02

**Summary:**

This paper develops a RED with modality-specific disentangled representation for motion deblurring. The main problems pointed out by reviewers include the higher computational efficiency, limited novelty, limited theoretical analysis of RPS and MRM, and poor writings.

**Reviewer Concerns:**

In the rebuttal, the authors provides results and corresponding analysis to solve the concerns of higher computational efficiency and limited theoretical analysis of RPS and MRM.

However, the novelty of the paper is limited. The novelty of the MRM, as well as MSEM and ESEM, is not  significant. The perturbation strategy looks like a data augmentation trick.

**Reviewer Scores:**

Although the authors partly solve the concerns of reviewers, the novelty of the paper is limited as pointed out by Reviewer Gsta and Reviewer bp3s.

---

### Decision · Program_Chairs · 2026-01-26

Reject